# A conserved phage phosphoesterase enables evasion of bacterial antiviral immunity

Junlong Li [ID][1], Yihao Song[1], Xiao Guo [ID][1] & Zheng-Guo He [ID][1,2]✉

## Abstract

**With the increasing prevalence of drug-resistant bacteria, anti-microbial resistance emerges as a global public health threat. Mycobacteriophages show exciting prospects for the treatment of drug-resistant bacterial infections. However, the molecular mechanism through which they escape host bacterial defenses remains unclear. Here, we report that the gene *gp48* of the mycobacteriophage A10ZJ24, which encodes a metallophosphoesterase-like protein, is essential for killing *Mycobacterium tuberculosis*. Gp48 is expressed during early stages of phage infection, and the Gp48 protein efficiently disrupts mycobacterial genomic DNA integrity, thereby silencing the expression of multiple anti-phage defense genes. While *gp48*-deletion phages infect and inject their DNA normally into *M. tuberculosis* cells, they are not able to impair the activation of the bacterial anti-phage genes which inhibit the replication of the genomic DNA of the mutant phage. This study thus identifies a phage metallophosphoesterase as a novel tool for subverting host bacterial antiviral immunity and killing *M. tuberculosis*. Our work fills a critical gap in the current knowledge on the arms race between mycobacteriophages and *M. tuberculosis*.**

**Keywords** *Mycobacterium tuberculosis*; Phage; Antiphage Defense; Metallophosphoesterase
**Subject Categories** Microbiology, Virology & Host Pathogen Interaction; Signal Transduction

## Introduction

With the increased prevalence of drug-resistant bacteria, anti-microbial resistance has become a global public health threat. As a large group of viruses that specifically infect bacteria (Dion et al, 2020; Feiner et al, 2015), bacteriophages or phages have shown promising prospects for the treatment of drug-resistant bacterial infections. However, when phages attack bacteria, they are inevitably constrained by bacterial anti-phage defense systems, such as the restriction and modification (RM) (Wilson & Murray, 1991), single-stranded DNA phosphorothioate modification (Ssp)

(Xiong et al, 2020), defense island system associated with restriction-modification (DISARM) (Ofir et al, 2018), clustered regularly interspaced short palindromic repeats (CRISPR-Cas) (Marraffini, 2015), bacteriophage exclusion (BREX) (Goldfarb et al, 2015), and abortion infection (Abi) (Lopatina et al, 2020) systems. Phages have also evolved new anti-defense strategies, such as anti-CRISPR proteins (Bondy-Denomy et al, 2013), anti-RM proteins (Rifat et al, 2008), and anti-CBASS proteins (Hobbs et al, 2022).

Mycobacteria are a large family of special actinomycetes that include *Mycobacterium tuberculosis*, *M. bovis* BCG, and non-tuberculosis strains such as *M. abscessus* and *M. smegmatis* (Eisenstadt and Hall, 1995; Koch and Mizrahi, 2018; Koh, 2017). *M. tuberculosis* represents a typical intracellular pathogen that causes of tuberculosis in humans, leading to more than 10 million new tuberculosis cases and almost 1.6 million deaths globally each year (Koch and Mizrahi, 2018). Strikingly, phages isolated using *M. smegmatis* as the host can effectively lyse *M. abscessus* and these phages have shown great potential for the clinical treatment of drug-resistant *M. abscessus* (Nick et al, 2022). However, only a small proportion of mycobacteriophages have been found to kill *M. tuberculosis* (Jacobs-Sera et al, 2012). In particular, the molecular mechanism underlying their successful escape from antiviral defense against *M. tuberculosis* is largely unclear, and the related anti-defense genes encoded by mycobacteriophages remain to be characterized.

In this study, we report a mycobacteriophage-encoded metallo-phosphoesterase (MPE) that can function as a novel anti-defense weapon to subvert the antiviral immunity of *M. tuberculosis*. Metallophosphoesterases, also known as calcineurin-like MPEs, are a superfamily of hydrolases containing bimetallic ion-binding centers that hydrolyze phosphate monoesters, diesters, or triesters (Matange et al, 2015). These include protein phosphatases of *E. coli* lambda phage (Barik, 1993), bacterial or archaeal cyclic nucleotide hydrolases and nucleotidases (Imamura et al, 1996), and purple acid phosphatases in plants (Bhadouria and Giri, 2022). Although these proteins have low sequence similarity, they all have five conserved motifs that form the active domain of MPE (Matange et al, 2015). However, different types of MPEs typically have different substrates and functions in different organisms (Barik, 1993; Imamura et al, 1996; Bhadouria and Giri, 2022; Ghosh et al, 2021; Paull, 2018). In mycobacterial Cluster A phages, the amino acid sequences of MPEs are highly conserved, but their substrates and functions are different. Gp47, encoded by the Bxb1 phage, does

[1]College of Life Science and Technology, Guangxi University, Nanning 530004, China. [2]State Key Laboratory of Virology, Taikang Center for Life and Medical Sciences, TaiKang Medical School (School of Basic Medical Sciences), Wuhan University, Wuhan 430071, China. ✉E-mail: hezhengguo2024@whu.edu.cn

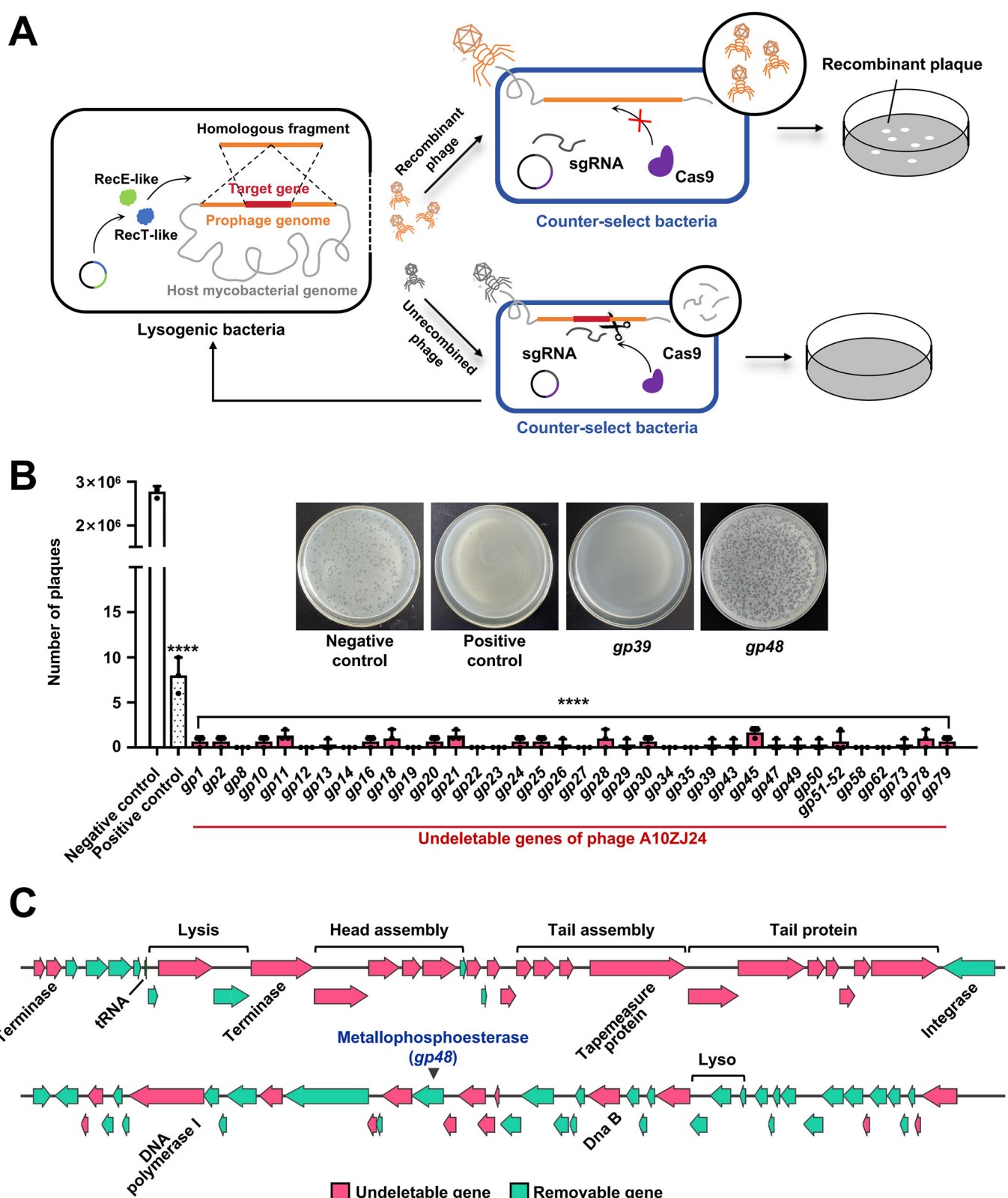

Figure 1. Gene essentiality assays of phage A10ZJ24 for infecting *Mycobacterium smegmatis*.

(A) Schematic diagram of the deletion of the phage gene using the bacteriophage recombineering of electroporated DNA (BRED) strategy in lysogen A10ZJ24. A target gene of the prophage A10ZJ24 was knocked out in lysogenic *M. smegmatis* and the mutant phage was isolated using the CRISPR-Cas9 counter-selection strain. DNA fragments containing upstream and downstream sequences of the target gene were transformed into lysogenic bacteria and then exchanged with homologous DNA within the prophage genome with the aid of RecE-like and RecT-like proteins. The genomes of the unrecombined wild-type phages were cut using Cas9 and eliminated using counter-selection bacteria. Recombinant plaques were picked and analyzed by polymerase chain reaction (PCR) to confirm whether the target gene had been knocked out. (B) Plaque-formation assays for gene deletion in A10ZJ24 using the BRED strategy. Plaque growth after knockout of *gp48* and an undeletable gene *gp39* are shown in the upper panel. Plaque-formation units on the lawns of counter-selected bacteria were used as positive controls, while the control buffer was used as a negative control. All data are presented as the mean ± SD ($n = 3$, biological replicates). The $P$ values were calculated by two-way ANOVA using GraphPad Prism v7.0 ($p < 0.0001$ for all undeletable genes and negative control versus positive control, ****$p < 0.0001$). (C) Map summarizing the deleted and undeleted genes of phage A10ZJ24 for infecting *M. smegmatis* mc² 155 obtained after BRED screening. Green arrows represent the deletable genes. Red arrows represent undeletable genes with no plaque formation on the lawn of the counter-selection strain or target genes that could not be knocked out by PCR after three rounds of screening experiments. The arrow indicates the transcriptional direction of the gene. Source data are available online for this figure.

not exhibit phosphoesterase activity and functions as a recombinant directional factor in the regulation of the lysogenic lysis process (Savinov et al, 2012). Gp66 encoded by the D29 phage can use monoesters, diesters, or cAMP as substrates, and the expression of Gp66 inhibits the growth of phages and *M. smegmatis* (Dutta et al, 2014).

In this study, we utilized a modified bacteriophage recombineering of electroporated DNA (BRED) strategy (Wetzel et al, 2021) to produce a series of mutant strains of phage A10ZJ24, which was isolated from a soil sample in China and belongs to Cluster A mycobacteriophages. By comparing the differential ability of these mutant phages to infect three mycobacterial strains, we found that the phage lacking *gp48* could lyse *M. smegmatis*, but lost the ability to kill *M. bovis* BCG and *M. tuberculosis*. This study identified a conserved MPE of Cluster A mycobacteriophages as a new weapon for subverting host bacterial antiviral immunity and uncovered a new mechanism for the killing of *M. tuberculosis* by mycobacteriophages.

## Results

### Gp48 is one of 42 phage genes not essential for infecting *M. smegmatis*

A10ZJ24 can infect and lyse multiple mycobacterial strains, including *M. smegmatis*, *M. bovis* BCG, and *M. tuberculosis*. To search for potential non-tail-structure proteins responsible for viral host range, a modified BRED method was first utilized to produce a series of single-gene-deleted mutant phages. As shown in Fig. 1A, a target gene of prophage A10ZJ24 was knocked out in lysogenic *M. smegmatis*, and the mutant phage was further isolated and purified using the CRISPR-Cas9 counter-select strain, in which the unrecombined wild-type phage could not survive and was eliminated. As shown in Fig. 1B, when no homologous DNA fragment for any target gene of the phage was transformed into the lysogenic strain, $(2.7 \pm 0.2) \times 10^6$ plaques grew in the double-layer plate containing the negative control strain without a counter-selection plasmid. In contrast, few plaques were observed in the positive control strain containing the counter-selection plasmid. This indicated that the screening system worked well. Using this strategy, we screened all genes encoded by the genome of phage A10ZJ24 (Appendix Fig. S1). Thirty-seven genes could not be

knocked out, and the resulting mutant phages were not produced, even after repeating the screening procedures three times (Fig. 1B).

Figure 1C summarizes our screening results. The deletable genes are indicated in green and the undeletable genes are indicated in red. Strikingly, 42 deletable or non-essential phage genes for infecting *M. smegmatis*, including the *gp48* gene, were successfully knocked out, and we successfully obtained single-gene-deleted mutant phages.

### The *gp48* gene is required for the killing of *M. tuberculosis* by phages

We compared the ability of these mutant phages to infect and lyse three different host mycobacterial strains. As shown in the upper panel of Fig. 2A, when all the mutant phages were spotted onto the medium, except for *gp30*, they formed clear plaques in the medium containing *M. smegmatis* (left panel). However, no plaques were observed for several mutant phages when they were spotted on medium containing BCG (middle panel), indicating a loss of lytic ability. These genes, which are important for BCG infection, are mainly located between *gp39* and *gp53* (Fig. 2A lower panel, Appendix Fig. S2). A further series of phage dilution experiments confirmed this observation. As shown in Fig. 2B, when continuously diluted 10-fold ($10^0$–$10^{-4}$), these mutant phages clearly lysed the mycobacteria and grew well on medium containing *M. smegmatis*. In contrast, only plaques on medium containing BCG were observed for these phages at relatively low dilutions of two or three (Fig. 2B, right panel), indicating that these genes significantly contributed to phage infection of BCG.

Strikingly, the *gp48*-deleted mutant phage almost lost its lytic ability when diluted ($10^0$–$10^{-4}$) and spotted onto medium containing *M. tuberculosis* (Fig. 2A,C). Even at the highest titer, the mutant phage had a very weak ability to infect *M. tuberculosis*. A similar essentiality of *gp48* was observed for the lytic ability of the mutant phage to infect BCG (Fig. 2B, right panel). These results indicated that Gp48 is required for A10ZJ24 infection of both BCG and *M. tuberculosis*.

### Gp48 is critical for ensuring the DNA replication of phages in *M. tuberculosis*

Next, we attempted to determine the potential mechanism by which the *gp48*-deleted mutant phage lost its ability to lyse *M.*

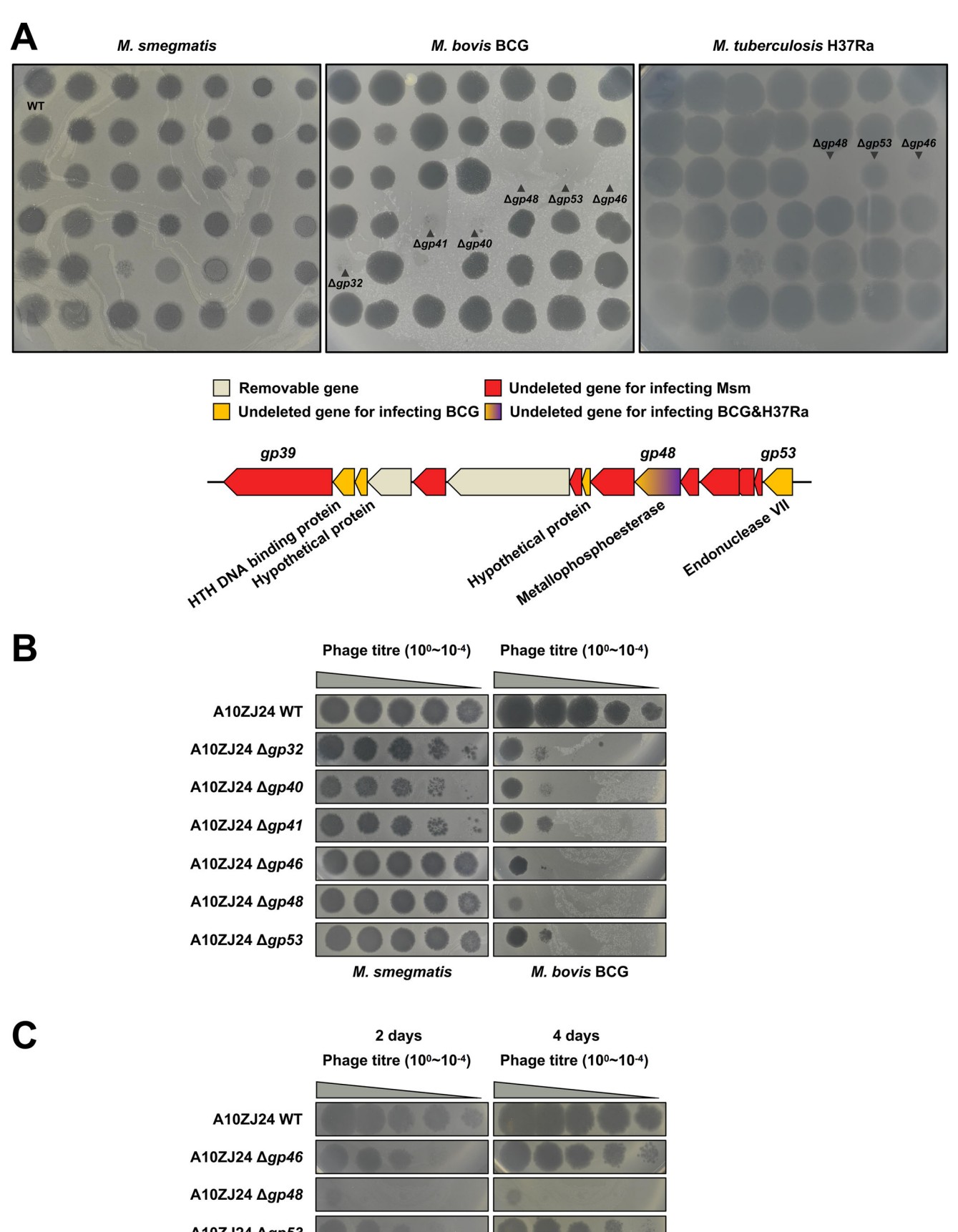

**Figure 2. Differential effects of *gp48* deletion on the infecting ability of A10ZJ24 for three different host mycobacteria.**

(A) Comparative assays of the infectivity of various mutant phages A10ZJ24. Wild-type A10ZJ24 and 41 different gene-deleted phages were spotted onto lawns of *M. smegmatis* mc$^2$ 155, *M. bovis* BCG, and *M. tuberculosis* H37Ra. The lower panel shows schematic diagrams of the gene essentiality of phage A10ZJ24 for infecting different hosts. (B) Comparative assays of the plaque-formation ability of several mutant phages on the lawns of *M. smegmatis* mc$^2$ 155 (left panel) and *M. bovis* BCG (right panel). Two microliters of 10-fold serial dilutions of the phages were spotted on the lawns of the strains. (C) Plaque-formation ability of the three mutant phages on the lawns of *M. tuberculosis* H37Ra. Plaque formation was assessed after incubating for 2 days and 4 days at 37 °C, respectively. Source data are available online for this figure.

*tuberculosis*. The first step in successful infection is the effective adsorption of phages onto the surface of the host bacteria. We first examined whether there was a differential absorption ability between the wild-type and A10ZJ24 mutant phages. As shown in Fig. 3A, when determining the adsorption of the wild-type and mutant phage to the surface of *M. tuberculosis* H37Ra at two time points, 30 min and 60 min after infection, we found no significant difference between the wild-type ($51.7 \pm 10\%$ at 60 min) and A10ZJ24 mutant ($45.6 \pm 10\%$ at 60 min) phages, indicating that *gp48* deletion did not affect phage adsorption on the cell surface of *M. tuberculosis*.

We then utilized quantitative reverse transcription polymerase chain reaction (qRT-PCR) assays to determine the expression levels of early-stage genes of the phage at three time points (4, 8, and 12 h) during the infection of *M. tuberculosis* and to observe whether the genomic DNA of the mutant phage was normally injected into the host mycobacteria. As shown in Fig. 3B, when wild-type phage A10ZJ24 infected *M. tuberculosis*, significant expression levels of an early-stage gene encoding DNA polymerase I, *gp39*, was observed 4 h after infection. The expression level was almost 15-fold higher at 4 h than at 0 h. Of note, an even greater change in expression level was observed for *gp48* 4 h after infection under similar conditions, with almost a 120-fold increase compared with the level at 0 h. This indicated that *gp48* is highly induced during the early stages of infection with phage A10ZJ24. In contrast, no significant differences in expression were observed between the lysogen-related gene *gp31* and the internal control mycobacterial *16S rRNA* gene. A similar expression level of *gp39* was observed after infection with the *gp48*-deleted mutant phage (Fig. 3C) and after infection with the wild-type phage under the same conditions, indicating that the mutant phage DNA was successfully injected into *M. tuberculosis*. Consistently, no significant expression of *gp48* was observed even 12 h after infection with the mutant phage (Fig. 3C). These results showed that the deletion of *gp48* did not affect the injection of the mutant phage DNA into the host mycobacteria.

Next, we used qPCR to examine whether *gp48* deletion would result in a defect in phage genomic DNA replication in the host *M. tuberculosis*. As shown in Fig. 3D, 18 h after infection, the relative DNA abundance of the mutant phage in the *M. tuberculosis* strain was only $0.33 \pm 0.01$, while the value in the wild-type strain was almost 10-fold higher ($3.35 \pm 0.04$). In contrast, no obvious difference in DNA abundance between the wild-type and mutant phages was observed during infection of another host, *M. smegmatis* (Appendix Fig. S3). These results indicated that the *gp48*-deleted mutant phage lost the ability to complete genomic DNA replication in *M. tuberculosis*, although it could normally infect and lyse *M. smegmatis*.

Taken together, these data indicate that *gp48* deletion did not affect the phage absorption or DNA injection processes, but Gp48

was critical for ensuring genomic DNA replication of phage A10ZJ24 in *M. tuberculosis*.

## Gp48 functions as a conserved MPE-like protein

Next, we confirmed that *gp48* encodes a MPE-like protein in the genome of A10ZJ24. As shown in Fig. 4A, the amino acids sequence of Gp48 is highly similar to that of other MPEs in Cluster A mycobacteriophages and contain all five conserved motifs of a typical MPE. To address this issue, we constructed several ATc-inducible expression strains carrying A10ZJ24 Gp48 mutants with single amino acid substitutions (Appendix Fig. S4A), and confirmed that several conserved motifs play critical roles in Gp48 function and in maintaining the *M. tuberculosis*–lysing ability of A10ZJ24. To further search for potentially functional homologs of Gp48 from Cluster A mycobacteriophages, we constructed several recombinant A10ZJ24 phages by utilizing A1SD1 *gp49*, A9GX2 *gp53*, and A3JS8 *gp53* to replace A10ZJ24 *gp48* in the genome of phage A10ZJ24. As shown in the right panel of Fig. 4B, both A9GX2 *gp53* and A3JS8 *gp53* could not replace A10ZJ24 *gp48* to produce clear plaques on the lawns of *M. tuberculosis* H37Ra. In contrast, the recombinant phage in which A10ZJ24 *gp48* was replaced by A1SD1 *gp49* formed clear plaques under the same conditions, and its growth phenotype was very similar to that of the wild-type phage A10ZJ24, suggesting that A1SD1*gp49* can functionally replace A10ZJ24 *gp48*.

We further confirmed the importance of the conserved motifs of A1SD1 *gp49* in the recombinant A10ZJ24 phage. A series of recombinant A10ZJ24 phages were constructed by replacing *gp48* in the genome of phage A10ZJ24 with A1SD1 *gp49* mutants containing a single amino acid mutation. As shown in Fig. 4C, the wild-type A1SD1 *gp49* gene restored the defective phenotype of A10ZJ24 Δ*gp48* and the recombinant phage re-obtained clear lytic ability for *M. tuberculosis*, which was similar to the phenotype of the wild-type phage. In contrast, all three recombinant phages, each containing a point mutation in the conserved motif of Gp49 (Fig. 4C, right panel), lost their ability to lyse *M. tuberculosis*. A1SD1 *gp49* (A113E), in which the mutated amino acid residue of Gp49 was located outside of the conserved motifs, was used as a negative control. As expected, the recombinant phage formed clear plaques on the lawns of *M. tuberculosis* H37Ra, which was very similar to the phenotype of the wild-type phage.

These results strongly suggested that A10ZJ24 *gp48* encodes an MPE-like protein. To confirm this hypothesis, we purified A10ZJ24 Gp48 and its mutant protein (Appendix Fig. S4B) and determined its phosphomonoesterase and phosphodiesterase activities. As shown in Fig. 4D, SAP, which was used as a positive control, represented alkaline phosphatase (shrimp) and had relatively high phosphomonoesterase activity (6765%), whereas its phosphodies-

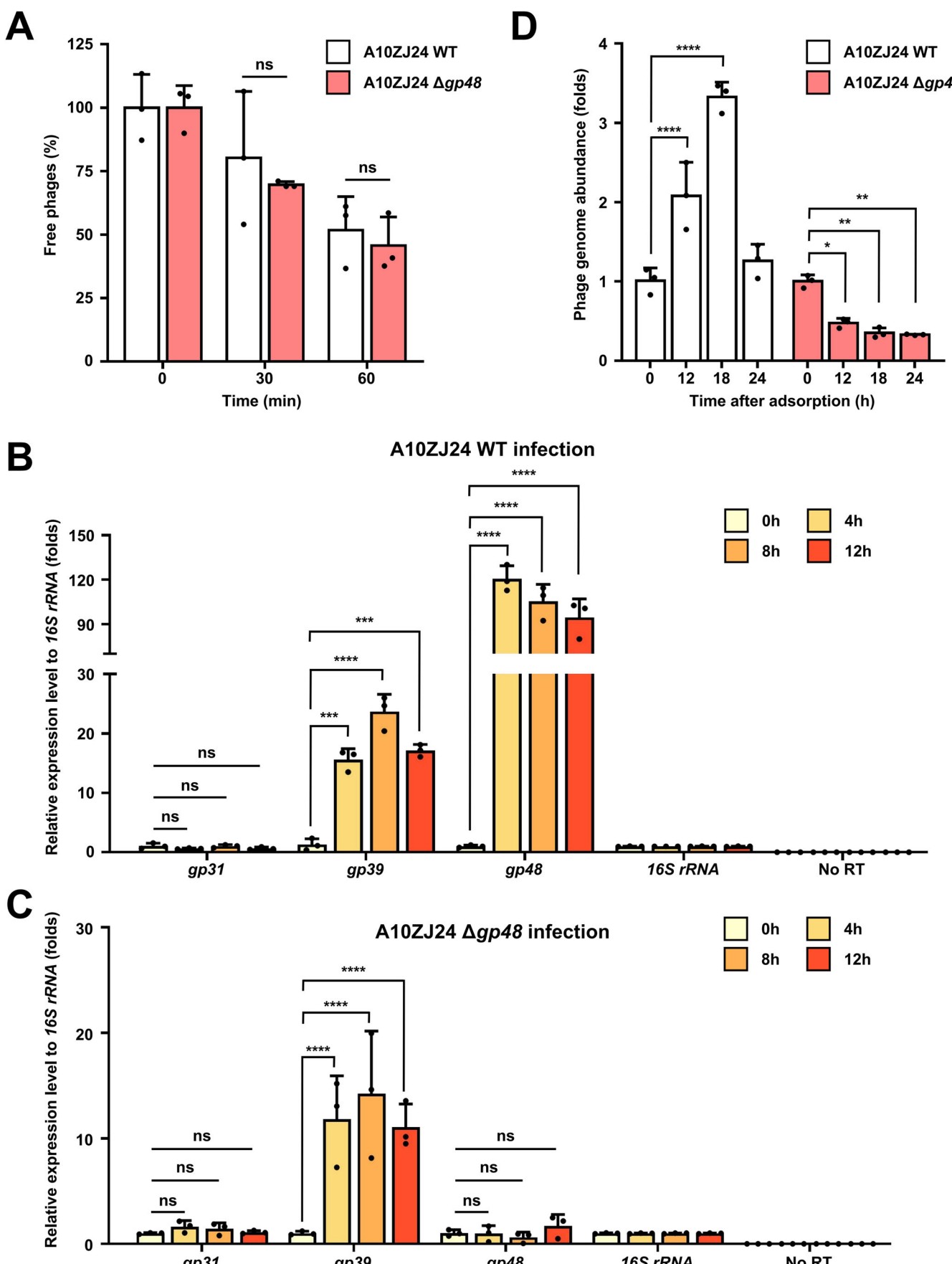

**Figure 3. Effect of *gp48* deletion on viral absorption, DNA injection, and genomic DNA replication in A10ZJ24-infected *M. tuberculosis*.**

(A) Comparative assays of the adsorption of the wild-type and mutant phage A10ZJ24 on the surface of *M. tuberculosis* H37Ra strain. (B) Quantitative reverse transcription (qRT) PCR assays for gene expression of the wild-type phage A10ZJ24 at several time points (0, 4, 8, and 12 h) after the phage entered *M. tuberculosis* H37Ra cells. Both *gp39* and *gp48* are early-stage-infection genes, and *gp31* is a lysogen-related gene. *16S rRNA* gene was used as an endogenous negative control. (C) qRT-PCR assays of gene expression of the mutant phage A10ZJ24 Δ*gp48* at several time points (0, 4, 8, and 12 h) after the phage entered *M. tuberculosis* H37Ra cells. (D) Quantitative PCR assays of the relative abundance of the phage genome at different time points after wild-type and mutant phage infection of *M. tuberculosis* H37Ra. All data are presented as the mean ± SD ($n = 3$, biological replicates). The $P$ values were calculated by two-way ANOVA using GraphPad Prism v7.0. Asterisks denote significant differences (ns = non-significant, \*$p < 0.05$, \*\*$p < 0.01$, \*\*\*$p < 0.001$, \*\*\*\*$p < 0.0001$) between two groups. (A) Free phages at 30 min: $p = 0.7683$, at 60 min: $p = 0.9417$; (B) The expression level of A10ZJ24 *gp31* at 4 h: $p = 0.9988$, at 8 h: $p > 0.9999$, at 12 h: $p = 0.9989$; Expression level of *gp39* at 4 h: $p = 0.0008$, at 8 h: $p < 0.0001$, at 12 h: $p = 0.0002$; Expression level of *gp48* at 4 h: $p < 0.0001$, at 8 h: $p < 0.0001$, at 12 h: $p < 0.0001$; (C) The expression level of A10ZJ24 mutant *gp31* at 4 h: $p = 0.9405$, at 8 h: $p = 0.9803$, at 12 h: $p = 0.9996$; Expression level of *gp39* at 4 h: $p < 0.0001$, at 8 h: $p < 0.0001$, at 12 h: $p < 0.0001$; Expression level of *gp48* at 4 h: $p > 0.9999$, at 8 h: $p = 0.9829$, at 12 h: $p = 0.9301$; (D) Genome abundance of A10ZJ24 at 12 h: $p < 0.0001$, at 18 h: $p < 0.0001$, at 24 h: $p = 0.2935$; Genome abundance of A10ZJ24 mutant at 12 h: $p = 0.0111$, at 18 h: $p = 0.0021$, at 24 h: $p = 0.0016$. Source data are available online for this figure.

terase activity was very weak (2.4%). Clear phosphodiesterase activity of A10ZJ24 Gp48 was observed, but its mutant protein, A10ZJ24 Gp48 (D10G), completely lost its activity under similar experimental conditions (Fig. 4D, right panel). The reaction buffer without protein, which was used as a negative control, did not show any activity. These results showed that A10ZJ24 Gp48 encodes an MPE.

Taken together, these results suggested that A10ZJ24 Gp48 functions as a conserved MPE-like protein in mycobacteriophages.

## Gp48 is toxic to host bacteria and disrupts mycobacterial genomic integrity

Our data showed that A10ZJ24 *gp48* encodes an MPE-like protein that is highly induced at an early stage of phage infection and the *gp48*-deleted mutant phage cannot normally replicate its genomic DNA in *M. tuberculosis*. To further pursue the potential mechanism whereby Gp48 confers phage A10ZJ24 with the ability to lyse *M. tuberculosis*, we determined the effect of Gp48 expression on the growth of the host mycobacteria. As shown in Fig. 5A and Appendix Fig. S5A, when a final concentration of 0.02 or 0.1 mg/mL ATc was added to the solid or liquid medium to induce gene expression, the growth inhibition of the recombinant *M. tuberculosis* H37Ra strain containing an ATc-induction-dependent expression plasmid, pJR962-A10ZJ24 *gp48*, was clearly observed both on the solid plate (Fig. 5A) and in the liquid medium (Appendix Fig. S5A). This indicated that Gp48 is a growth inhibitor and is toxic to *M. tuberculosis*. Using a similar strategy, we evaluated the importance of several conserved motifs for the toxicity of Gp48 against the host mycobacteria. When the three mutant proteins were induced with 0.1 mg/mL ATc in liquid medium, Gp48 (D10G), Gp48 (D41V), and Gp48 (H175D) lost their ability to inhibit the growth of both *M. tuberculosis* (Fig. 5B) and *M. smegmatis* (Appendix Fig. S5C), indicating that these residues are important for the toxicity of Gp48. In contrast, under similar experimental conditions, the mutant protein Gp48 (A113A), in which the mutated amino acid residue is outside of the conserved motifs, still significantly inhibited the growth of both host mycobacteria upon induction with 0.1 mg/mL ATc. Strikingly, 0.01 mg/mL ATc induced the death of *M. smegmatis*, but 0.1 mg/mL ATc was required to induce the death of *M. tuberculosis* under similar experimental conditions (Appendix Fig. S5A,B). This result is consistent with our observation that conserved motifs are

essential for phage infection of *M. tuberculosis* (Appendix Fig. S4A; Fig. 4C).

As some MPEs have been shown to possess nuclease activity and participate in bacterial DNA damage repair (Matange et al, 2015; Ghosh et al, 2021; Paull, 2018), we analyzed the potential effect of Gp48 expression on the genomic DNA integrity of the host mycobacteria in vivo. *M. tuberculosis* H37Ra containing an ATc-induction-dependent plasmid, with or without the *gp48* gene, was grown in 7H9 medium. As shown in Fig. 5C, when a final concentration of 0.1 mg/mL ATc was added to the medium for induction, the content of complete high-molecular-weight genomic DNA in the strain containing the plasmid pJR962-*gp48* decreased in a gradient manner as the induction time increased from 12 to 24 h. In contrast, no obvious change was observed in the strain containing the empty plasmid or the strain containing pJR962-*gp48* (D10G) under the same induction conditions. Similar results were observed when the expression of the *gp48* gene was induced in *M. smegmatis* (Appendix Fig. S5D). Furthermore, we determined the in vitro nuclease activity of purified MPE and its mutant proteins. As shown in Fig. 5D, a time-course experiment (from 1 to 60 min) revealed that linear DNA, circular plasmids, and mycobacterial genomes were gradually degraded by the wild-type MPE protein. Conversely, no nuclease activity was observed for the mutant proteins under similar conditions.

These results indicate that the expression of A10ZJ24 *gp48* is toxic to the host bacteria and that Gp48 disrupts mycobacterial genomic integrity.

## Gp48 plays a counter-defensive role against the defense mechanism of *M. tuberculosis*

Genomic integrity disruption induced by Gp48 would significantly modify the gene expression profile of *M. tuberculosis*, particularly in some unknown anti-phage genes, which would ultimately affects the outcome of the phage-mycobacterium interaction. To confirm this assumption and search for potentially upregulated anti-phage genes upon infection with the mutant phage, we utilized RNA sequencing assays to determine the difference in gene expression levels in the *M. tuberculosis* H37Ra strain at 1 h and 10 h post-infection with wild-type phage A10ZJ24 or its Δ*gp48* mutant. As shown in Fig. 6A, 668 genes had significant expression level differences (log2 fold change ≥0.5 and $P$-value ≤ 0.05) in *M. tuberculosis* between 1 h and 10 h after mutant phage infection.

Strikingly, 37 upregulated genes were identified in *M. tuberculosis* 10 h after mutant phage infection, and these genes had no homologs in the genome of *M. smegmatis*. Of note, these 37 genes were not upregulated after infection with wild-type A10ZJ24 (Fig. 6B).

Next, we used an ATc-dependent expression plasmid to examine whether these upregulated genes have anti-phage ability. As shown in Fig. 6C, the expression of certain genes, such as *Mra_1649* and *Mra_3122*, increased the anti-A10ZJ24 ability of *M. tuberculosis* H37Ra by at least 100-fold when a final concentration of 0.1 mg/mL ATc was added to the medium for induction, indicating that these genes have clear anti-phage functions. Similarly, the expression of *Mra_1649*, *Mra_1940A*, *Mra_2329A*, and *Mra_3122* also conferred more than 100-fold phage resistance on *M. smegmatis* compared to the wild-type strain (Appendix Fig. S6A). qRT-PCR assays confirmed that these four genes were upregulated upon infection with A10ZJ24 Δ*gp48*, but not upon infection with the wild-type phage (Appendix Fig. S6B). Consistently, when the CRISPRi technique was used to inhibit the expression of *Mra_1649* or *Mra_3122* in *M. tuberculosis* H37Ra, the plaque-formation capacity of the A10ZJ24 *gp48*-deletion phage was partially restored in the lawn of recombinant *M. tuberculosis* (Fig. 6D).

*Mra_1649*, *Mra_1940A*, *Mra_2329A*, and *Mra_3122* play widespread defensive roles against different types of mycobacteriophages. As shown in Fig. 7A and Appendix Fig. S7A, when these four genes were expressed in *M. smegmatis* after induction with ATc, the recombinant mycobacterium showed obvious defense ability against multiple types of mycobacteriophages, including clusters A2, A3, A4, K1, K3, K4, K5, and L1, when compared with the strain without the induction with ATc. A similar finding was observed in *M. tuberculosis* H37Ra cells. As shown in Fig. 7B and Appendix Fig. S7B, the expression of *Mra_1649*, *Mra_1940A*, or *Mra_2329A* in *M. tuberculosis* endowed the recombinant mycobacterium with significant defense against different clusters of mycobacteriophages. Strikingly, the expression of *Mra_3122*, which encodes a protease, resulted in clear resistance of *M. tuberculosis* to infection with phages A3, A4, K4, and K5.

Taken together, these data indicate that Gp48 plays a counter-defensive role in the anti-phage defense of *M. tuberculosis*. A10ZJ24 Δ*gp48* infection activated expression of multiple anti-phage genes of *M. tuberculosis* which have widespread defensive functions against infection with different types of mycobacteriophages, and these genes have no homologs in the genome of *M. smegmatis*.

## Discussion

We identified the *gp48* gene of the phage A10ZJ24, which encodes a conserved MPE in multiple mycobacteriophages, as a new tool for subverting host bacterial antiviral immunity. The *gp48*-deleted mutant phage was able to infect and lyse *M. smegmatis*, but lost the ability to kill *M. tuberculosis*. Our data support the model shown in Fig. 7C. When phage A10ZJ24 infected *M. tuberculosis*, the *gp48* gene was expressed, and bacterial genomic DNA integrity was disrupted by its phosphodiesterase activity. Therefore, the anti-phage genes of *M. tuberculosis* cannot be normally activated to inhibit phage genome replication and progeny phages can be produced. Thus, we discovered a new counter-defense strategy for

mycobacteriophages to survive in *M. tuberculosis* by silencing the expression of multiple anti-phage genes of the human pathogen.

We utilized an improved BRED gene knockout strategy to produce 42 mutant strains of phage A10ZJ24 with a single-gene knockout, which provided us with the opportunity to discover potential genes that determine the host range of phage infection. Recently, Cas proteins with nucleic acid cleavage activity have been used to develop new strategies for deleting phage genes. For example, Yuan et al used the Cas9 protein to perform gene mutation and knockout of four *E. coli* phages and named it CRISPR/Cas9-based iterative phage genome reduction (CiPGr) (Yuan et al, 2022). Adler et al combined CRISPR-Cas13a with homologous recombination to perform gene knockout and single-base editing in the T4, T7, and EdH4 phages (Adler et al, 2022). For mycobacteriophages, Marinelli et al developed a homologous recombination-based BRED technique for gene deletion using RecE/RecT-like Che9 protein (Marinelli et al, 2008). Dedrick et al used this method to successfully identify 35 non-essential and 19 essential genes in phage Giles (Dedrick et al, 2013). A phage counter-selection strategy based on CRISPR-Cas9 was developed to improve the identification efficiency of positive recombinant phages (Wetzel et al, 2021). In the present study, a lysogenic strain was first produced by infecting *M. smegmatis* with a high titer of phage A10ZJ24, and phage genomic DNA was successfully integrated into the host genome containing a RecE/RecT-like protein plasmid. This greatly improved the efficiency of gene recombination and mutant phage screening when the upstream and downstream homologous fragments of the target gene were subsequently transformed into the lysogenic strain. We obtained 42 mutant phages in which a single non-essential gene was deleted, laying the foundation for subsequent studies of the essentiality of these genes for infecting different hosts. This strategy for improving recombination efficiency by constructing a lysogenic strain is not applicable to virulent phages.

A notable finding of this study is that a phage-encoded MPE endowed the phage with the ability to infect *M. tuberculosis*, thereby extending the host range of phage infection. There are very few studies on the host range of mycobacteriophages and there are few examples of phage structural proteins. Phage A10ZJ24 can widely infect and lyse various host mycobacteria, including *M. smegmatis*, *M. bovis* BCG, and *M. tuberculosis*. However, the genes that determine the host range and the underlying mechanisms are still unknown. In the present study, we identified a conserved MPE from phage A10ZJ24, encoded by the gene *gp48*, which is not required for infecting *M. smegmatis*, but is critical for killing *M. tuberculosis*. Therefore, Gp48 extends the host range of phage A10ZJ24 to *M. tuberculosis*. The MPE homolog of A10ZJ24, Gp48, is highly expressed in the early stages of phage infection, and its nuclease activity disrupts the genomic integrity of *M. tuberculosis*, thereby inhibiting the host's resistance to phage infection and allowing the phage to successfully replicate and survive. Therefore, A10ZJ24 confers the ability to infect *M. tuberculosis*.

Another important finding of this study was the identification of multiple novel genes with significant anti-phage roles in *M. tuberculosis*. To date, most studies on mycobacteriophage infection and host defense have been performed using fast-growing *M. smegmatis* as a model strain (Zhang et al, 2022; LeRoux et al, 2022; Seniya and Jain 2022; Li et al, 2023; Georjon and Bernheim, 2023).

However, there are few reports on phage infection and host defense mechanisms in slow-growing *M. tuberculosis*, and the potential anti-phage genes of *M. tuberculosis* remain unclear. Based on the finding that phages lacking the *MPE* gene have lost their ability to infect *M. tuberculosis*, we identified several genes with significant anti-phage defense functions, including *Mra_1649, Mra_2329A, Mra_3122*. Of note, these newly discovered anti-phage genes do not have homologs in the genome of *M. smegmatis* and are unique to *M. tuberculosis*. Moreover, when these genes were expressed in *M. smegmatis*, the recombinant mycobacterium showed significant resistance to a wide range of phages. Therefore, although unable to replicate successfully in *M. tuberculosis*, a mutant phage lacking MPE could effectively infect *M. smegmatis*.

In conclusion, we have successfully identified a conserved MPE of mycobacteriophages as a new tool for subverting host bacterial antiviral immunity and uncovered a previously unknown mechanism for the killing of *M. tuberculosis* by mycobacteriophages. This study fills a major gap in our current knowledge of the arms race between mycobacterial viruses and *M. tuberculosis*.

# Methods

### Reagents and tools table

| Reagent/Resource | Reference or Source | Identifier or Catalog Number |
| --- | --- | --- |
| **Experimental models** | | |
| *M. smegmatis* mc² 155 | ATCC | 700084 |
| *M. bovis* BCG | ATCC | 35734 |
| *M. tuberculosis* H37Ra | ATCC | 25177 |
| *E.coli* ArcticExpress (DE3) pRARE2 | WEIDI | EC2021 |
| A1SD1 | This study | PQ130132 |
| A3JS8 | This study | PQ130133 |
| A9GX2 | This study | PQ130134 |
| A10ZJ24 | This study | PQ130131 |
| A10ZJ24 Δ*gp32* | This study | N/A |
| A10ZJ24 Δ*gp40* | This study | N/A |
| A10ZJ24 Δ*gp41* | This study | N/A |
| A10ZJ24 Δ*gp46* | This study | N/A |
| A10ZJ24 Δ*gp48* | This study | N/A |
| A10ZJ24 Δ*gp53* | This study | N/A |
| A10ZJ24 Δ*gp48*:: A1SD1 *gp49* | This study | N/A |
| A10ZJ24 Δ*gp48*:: A3JS8 *gp53* | This study | N/A |
| A10ZJ24 Δ*gp48*::A9GX2 *gp53* | This study | N/A |
| A10ZJ24 Δ*gp48*:: A1SD1 *gp49* (D10G) | This study | N/A |
| A10ZJ24 Δ*gp48*:: A1SD1 *gp49* (D41V) | This study | N/A |
| A10ZJ24 Δ*gp48*:: A1SD1 *gp49* (A113E) | This study | N/A |
| A10ZJ24 Δ*gp48*:: A1SD1 *gp49* (H176D) | This study | N/A |
| **Recombinant DNA** | | |
| pET28a-SUMO | This study | N/A |
| pET-SUMO-A10ZJ24 *gp48* | This study | N/A |

| Reagent/Resource | Reference or Source | Identifier or Catalog Number |
| --- | --- | --- |
| pET-SUMO-A10ZJ24 *gp48* (D10G) | This study | N/A |
| pLJR962 | Addgene | #115162 |
| pLJR962 (Cas9) | This study | N/A |
| pLJR962 (Cas9)-A10ZJ24 *gp32* | This study | N/A |
| pLJR962 (Cas9)-A10ZJ24 *gp40* | This study | N/A |
| pLJR962 (Cas9)-A10ZJ24 *gp41* | This study | N/A |
| pLJR962 (Cas9)-A10ZJ24 *gp46* | This study | N/A |
| pLJR962 (Cas9)-A10ZJ24 *gp48* | This study | N/A |
| pLJR962 (Cas9)-A10ZJ24 *gp53* | This study | N/A |
| pJR962 | This study | N/A |
| pJR962-*gp48* | This study | N/A |
| pJR962-A10ZJ24 *gp48* (D10G) | This study | N/A |
| pJR962-A10ZJ24 *gp48* (D41V) | This study | N/A |
| pJR962-A10ZJ24 *gp48* (A113E) | This study | N/A |
| pJR962-A10ZJ24 *gp48* (H175D) | This study | N/A |
| pJR962-A10ZJ24 *gp48*-flag | This study | N/A |
| pJR962-A10ZJ24 *gp48*-flag (D10G) | This study | N/A |
| pJR962-A10ZJ24 *gp48*-flag (D41V) | This study | N/A |
| pJR962-A10ZJ24 *gp48*-flag (A113E) | This study | N/A |
| pJR962-A10ZJ24 *gp48*-flag (H175D) | This study | N/A |
| pJR962-*Mra_0950* | This study | N/A |
| pJR962-*Mra_1125* | This study | N/A |
| pJR962-*Mra_1160* | This study | N/A |
| pJR962-*Mra_1598A* | This study | N/A |
| pJR962-*Mra_1649* | This study | N/A |
| pJR962-*Mra_1685* | This study | N/A |
| pJR962-*Mra_1940A* | This study | N/A |
| pJR962-*Mra_1960* | This study | N/A |
| pJR962-*Mra_2459* | This study | N/A |
| pJR962-*Mra_2329A* | This study | N/A |
| pJR962-*Mra_2538* | This study | N/A |
| pJR962-*Mra_3122* | This study | N/A |
| pJR962-*Mra_3464* | This study | N/A |
| pJR962-*Mra_3701* | This study | N/A |
| pJR962-Mra IS6110 | This study | N/A |
| pJR962-Mra Csm | This study | N/A |
| pLJR965 | Addgene | #115163 |
| pLJR965-*Mra1649* | This study | N/A |
| pLJR965-*Mra1940A* | This study | N/A |
| pLJR965-*Mra2329A* | This study | N/A |
| pLJR965-*Mra3122* | This study | N/A |
| **Antibodies** | | |
| Anti-Flag tag mouse monoclonal antibody | Sangon Biotech | D191041-0100 RRID:AB 2940947 |
| HRP-conjugated Goat Anti-Mouse IgG | Sangon Biotech | D110087-0100 RRID: AB 2940948 |
| KatG rabbit polyclonal antibody | Li et al (2023) | N/A |

| Reagent/Resource | Reference or Source | Identifier or Catalog Number |
|---|---|---|
| HRP-conjugated Goat Anti-Rabbit IgG | Sangon Biotech | D110058-0100 RRID:AB 2940954 |
| **Oligonucleotides and other sequence-based reagents** | | |
| Primer name | Sequence (5'-3') | |
| A10ZJ24 *gp32* guide | F: 5'-GGGACCCTGTTTACCAGGGGTTTT-3' | |
| | R: 5'-AAACAAAACCCCTGGTAAACAGGG-3' | |
| A10ZJ24 *gp40* guide | F: 5'-GGGAGAGCGAGTCGTTCTCGTCGT-3' | |
| | R: 5'-AAACACGACGAGAACGACTCGCTC-3' | |
| A10ZJ24 *gp41* guide | F: 5'-GGGAGCCGGATTCGGTGAAGACCT-3' | |
| | R: 5'-AAACAGGTCTTCACCGAATCCGGC-3' | |
| A10ZJ24 *gp46* guide | F: 5'-GGGATAACCGCTGAGGCCGGCTTC-3' | |
| | R: 5'-AAACGAAGCCGGCCTCAGCGGTTA-3' | |
| A10ZJ24 *gp48* guide | F: 5'-GGGATTGTTCCAGCGGGACGGTTG-3' | |
| | R: 5'-AAACCAACCGTCCCGCTGGAACAA-3' | |
| A10ZJ24 *gp53* guide | F: 5'-GGGAGTTCCGGTTGCACGCGGTGC-3' | |
| | R: 5'-AAACGCACCGCGTGCAACCGGAAC-3' | |
| *Mra_1649* guide | F: 5'-GGGATGTCGTACCGCGGATCGGAC-3' | |
| | R: 5'-AAACGTCCGATCCGCGGTACGACA-3' | |
| *Mra_1940A* guide | F: 5'-GGGAAGCGCCGAATGCCTTGAGCG-3' | |
| | R: 5'-AAACCGCTCAAGGCATTCGGCGCT-3' | |
| *Mra_2329A* guide | F: 5'-GGGACGCCGCCGCTCATGGGCCG-3' | |
| | R: 5'-AAACTCGGCCCATGAGCGGCGGCG-3' | |
| *Mra_3122* guide | F: 5'-GGGATTGTCGAAGGTCTTGTACTG-3' | |
| | R: 5'-AAACCAGTACAAGACCTTCGACAA-3' | |
| A10ZJ24 *gp31* RT | F: 5'-AGAGAACTTCCTAGACCGCG-3' | |
| | R: 5'-CTGACTCAAGCTCAGCGATC-3' | |
| A10ZJ24 *gp39* RT | F: 5'-ATCCCAGAGGCAAAGACGAA-3' | |
| | R: 5'-CCCGAGTAGAGCTGGTAGTG-3' | |
| A10ZJ24 *gp48* RT | F: 5'-ACACCCAGATCCCTTACGAC-3' | |
| | R: 5'-CTGCATCCTTGAACACCGAG-3' | |
| *16S rRNA* RT | F: 5'-TGCTACAATGGCCGGTACAAA-3' | |
| | R: 5'-CGATTACTAGCGACTCCGACTT-3' | |
| *Mra_1649* RT | F: 5'-GACACCACCGGAAGCCAC-3' | |
| | R: 5'-GCTGCGCGTACTGATCTTC-3' | |
| *Mra_1940A* RT | F: 5'-GTACGAGGTGCTGCGAAAC-3' | |
| | R: 5'-CAATGTTGTCCTGGTGCACA-3' | |
| *Mra_2329A* RT | F: 5'-AGTAGCGACGACATCACCAT-3' | |
| | R: 5'-GATACCCTGAGCCCCGATAC-3' | |
| *Mra_3122* RT | F: 5'-AATGTGCCCACCATCCAGTA-3' | |
| | R: 5'-GATTCCCTTGTTGATCGCGG-3' | |
| pA10ZJ24 *gp48* | F: 5'-CTCACAGAGAACAGATTGGTGTGACCGAGCGCATCGTCGT-3' | |
| | R: 5'-TGGTGCTCGAGTGCGGCCGCTCAGACCTCCCAAGTGTGACCGTCT-3' | |
| pA10ZJ24 *gp48* (D10G) | F: 5'-CTCACAGAGAACAGATTGGTGTGACCGAGCGCATCGTCGTCATCAGCGGC-3' | |
| | R: 5'-TGGTGCTCGAGTGCGGCCGCTCAGACCTCCCAAGTGTGACCGTCT-3' | |
| A10ZJ24 *gp48* | F: 5'-GCGGCCGCAAGAATTCGTGACCGAGCGCATCGTCGT-3' | |
| | R: 5'-GCTTCTCGAGTCTAGATCAGACCTCCCAAGTGTGACCGTCT-3' | |

| Reagent/Resource | Reference or Source | Identifier or Catalog Number |
|---|---|---|
| A10ZJ24 *gp48* (D10G) | | F: 5'-GCGGCCGCAAGAATTCGTGACCGAGCGCATCGTCGTCATCAGCGGC-3' |
| | | R: 5'-GCTTCTCGAGTCTAGATCAGACCTCCCAAGTGTGACCGTCT-3' |
| A10ZJ24 *gp48* (D41V) | | F: 5'-CCACATCGGTGTCCTGATGGACTTCCCGCA-3' |
| | | R: 5'-CCATCAGGACACCGATGTGGATGACCCTGG-3' |
| A10ZJ24 *gp48* (A113E) | | F: 5'-CGAGTCTCGGGAATTTCACTTCG-3' |
| | | R: 5'-CGAAGTGAAATTCCCGAGACTCG-3' |
| A10ZJ24 *gp48* (H175D) | | F: 5'-CGTCATGGGCGACACCCACCGGCTGGGCAT-3' |
| | | R: 5'-GGTGGGTGTCGCCCATGACGACCGAGGTCC-3' |
| *Mra_0950* | | F: 5'-ATAGAGAAGGCGGTATCGATGCCTCGCTGACGCCGGCCAC-3' |
| | | R: 5'-CTTAGCTAATCAGCGGCCGCTCAAGTGGTGGCCGAGGCTG-3' |
| *Mra_1125* | | F: 5'-ATAGAGAAGGCGGTATCGATGTGATAAGCACCACAAGAAT-3' |
| | | R: 5'-CTTAGCTAATCAGCGGCCGCCTATTTCCCGACGATTACTG-3' |
| *Mra_1160* | | F: 5'-ATAGAGAAGGCGGTATCGATATGAGGGCAAGCCCGGCAGA-3' |
| | | R: 5'-CTTAGCTAATCAGCGGCCGCCTAGGGCGTGTCTCCCAATT-3' |
| *Mra_1598A* | | F: 5'-ATAGAGAAGGCGGTATCGATATGTTAGCGAATAGCCGGGA-3' |
| | | R: 5'-CTTAGCTAATCAGCGGCCGCTCAAAGGTGGCCCGCGCTTG-3' |
| *Mra_1649* | | F: 5'-ATAGAGAAGGCGGTATCGATATGCCCGACGAACCGACACC-3' |
| | | R: 5'-CTTAGCTAATCAGCGGCCGCTCACTCATCGGTGTGCATCG-3' |
| *Mra_1685* | | F: 5'-ATAGAGAAGGCGGTATCGATATGTCATTCCGACACGCTGC-3' |
| | | R: 5'-CTTAGCTAATCAGCGGCCGCTCAATGACACAGCGCCGGTC-3' |
| *Mra_1940A* | | F: 5'-ATAGAGAAGGCGGTATCGATACGAATGCGATTTGCGTCAT-3' |
| | | R: 5'-CTTAGCTAATCAGCGGCCGCCTACGGGCGACGTGCCTGCG-3' |
| *Mra_1960* | | F: 5'-ATAGAGAAGGCGGTATCGATTTGCTGCCGACCCTGTCGCA-3' |
| | | R: 5'-CTTAGCTAATCAGCGGCCGCCTATCTGTAGGTCAGCGCCG-3' |
| *Mra_2329A* | | F: 5'-ATAGAGAAGGCGGTATCGATTTGGCCACCAGTAGCGACGA-3' |
| | | R: 5'-CTTAGCTAATCAGCGGCCGCTCATGGGCCGACAGGAAGCT-3' |
| *Mra_2459* | | F: 5'-ATAGAGAAGGCGGTATCGATGTGGGCCTGCGCGACGCTGA-3' |
| | | R: 5'-CTTAGCTAATCAGCGGCCGCTCATTGTGCTGATGGATCAG-3' |
| *Mra_2538* | | F: 5'-ATAGAGAAGGCGGTATCGATATGACCTCTTCTCATCTTAT-3' |
| | | R: 5'-CTTAGCTAATCAGCGGCCGCCTAGGTGGTCAGTGCTGGGG-3' |
| *Mra_3122* | | F: 5'-ATAGAGAAGGCGGTATCGATATGACATGGCAGATCGTGTT-3' |
| | | R: 5'-CTTAGCTAATCAGCGGCCGCTTACCGTCCCGGCACTGCGA-3' |

| Reagent/Resource | Reference or Source | Identifier or Catalog Number |
|---|---|---|
| *Mra_3464* | | F: 5'-ATAGAGAAGGCGGTATCGATATGCCT AACCCGGTGACGAT-3' |
| | | R: 5'-CTTAGCTAATCAGCGGCCGCTCACTG CACGGGTATGGGCT-3' |
| *Mra_3701* | | F: 5'-ATAGAGAAGGCGGTATCGATGTGACC GTTAGCTACCACGC-3' |
| | | R: 5'-CTTAGCTAATCAGCGGCCGCTTATCG GCGCGGCGGATCGC-3' |
| Mra IS6110 | | F: 5'-ATAGAGAAGGCGGTATCGATTTGCGG TGGGGTGTCGAGTC-3' |
| | | R: 5'-CTTAGCTAATCAGCGGCCGCTCAGCC GGCGGCTGGTCTCT-3' |
| Mra Csm | | F: 5'-ATAGAGAAGGCGGTATCGATATAGGC GAGAACAGGATCAT-3' |
| | | R: 5'-CTTAGCTAATCAGCGGCCGCTCAAAA GAACACAAACTCCT-3' |
| A10ZJ24 *gp48*-flag | | F: 5'-CAGTGATAGAGAAGGCGGTATCGAT GTGACCGAGCGCATCGTCGTCATCA-3' |
| | | R: 5'-TCCTTGTAGTCGCCGCTGCTACCACC ACCACCGACCTCCCAAGTGTGACCGTCT-3' |
| Replace *gp48* Up | | F: 5'-GCTGGATCGTCGTTGCGATC-3' |
| | | R: 5'-TCACCGATCCCCCAGATCGA-3' |
| Replace *gp48* Dn | | F: 5'-TACTTGACATCGCTCAACGAGCTTG-3' |
| | | R: 5'-TTCCCAGGACATCTGGGCGT-3' |
| A1SD1 *gp49* | | F: 5'-TCGATCTGGGGGATCGGTGAATGACT AAACGAATCGCAAT-3' |
| | | R: 5'-TCGTTGAGCGATGTCAAGTATCAGAC TTCCCAGACCCGGC-3' |
| A3JS8 *gp53* | | F: 5'-TCGATCTGGGGGATCGGTGAGTGAG CAAGCGCATCGTCGT-3' |
| | | R: 5'-TCGTTGAGCGATGTCAAGTATCACAC CTCCCAGACTCGCC-3' |
| A9GX2 *gp53* | | F: 5'-TCGATCTGGGGGATCGGTGAGTGACC AAGCGAATCGTCAT-3' |
| | | R: 5'-TCGTTGAGCGATGTCAAGTACTAGAC CTTCCAGATGTTGC-3' |
| A1SD1 *gp49* (D10G) | | F: 5'-TCGATCTGGGGGATCGGTGAATGACT AAACGAATCGCAATTCTGCCCGGC-3' |
| | | R: 5'-TCGTTGAGCGATGTCAAGTATCAGAC TTCCCAGACCCGGC-3' |
| A1SD1 *gp49* (D41V) | | F: 5'-GGGAATCGGGGTTTACATGGACT-3' |
| | | R: 5'-AGTCCATGTAAACCCCGATTCCC-3' |
| A1SD1 *gp49* (A113E) | | F: 5'-GATGGAGGACGAATTCCACTTCC-3' |
| | | R: 5'-GGAAGTGGAATTCGTCCTCCATC-3' |
| A1SD1 *gp49* (H176D) | | F: 5'-GGTCATGGGCGACACCCATCGCA-3' |
| | | R: 5'-TGCGATGGGTGTCGCCCATGACC-3' |
| A10ZJ24 cos site | | F: 5'-CTCTGGTATCGCGGATAGGG-3' |
| | | R: 5'-GGGTATCGCTGCAGGTAGG-3' |
| **Chemicals, Enzymes and other reagents** | | |
| DNAse I | Takara | 2270A |
| OADC | BD | 211886 |
| PrimeSTAR® GXL DNA Polymerase | Takara | R050A |
| Restriction endonucleases (various) | Takara | N/A |
| Disodium p-nitrophenyl phosphate (pNPP) | Aladdin | P109039 |
| Bis(4-nitrophenyl) phosphate (bis-pNPP) | Aladdin | B113674 |

| Reagent/Resource | Reference or Source | Identifier or Catalog Number |
|---|---|---|
| Anhydrotetracycline hydrochloride | abcam | ab145350 |
| Alkaline Phosphatase (Shrimp) | Takara | 2660A |
| Acetamide | Sangon Biotech | A500687 |
| Disodium Succinate Hexahydrate | Sangon Biotech | A620889 |
| RN43-EASYspin Plus Bacteria/Tissue Cell RNA Rapid Extraction Kit | Aidlab | RN4302 |
| cDNA Reverse Transcription Kit | Aidlab | PC1803 |
| SYBR | Aidlab | PC3302 |
| Uniclone One Step Seamless Cloning Kit | Genesand | SC612 |
| Bacterial genomic DNA rapid extraction kit | Aidlab | DN11 |
| Western ECL Substrate | Sangon Biotech | C500044-0100 |
| **Software** | | |
| Image Lab v6.1 | BioRad | https://www.bio-rad.com/en-us/product/image-labsoftware?ID=KRE6P5E8Z |
| Snapgene v4.2.4 | Dotmatics | https://www.snapgene.com/ |
| GraphPad Prism v7.0 | Dotmatics | https://www.graphpad.com/ |
| **Other** | | |
| Applied Biosystems QuantStudio 3 | Thermo Fisher | |
| ChemiDoc™ XRS+ System | BIO-RAD | |
| Trans-Blot® Turbo™ Transfer System | BIO-RAD | |
| Tecan Infinite Spark | Tecan | |

## Bacterial strains, mycobacteriophages, and culture conditions

Bacterial strains and mycobacteriophages used in this study are listed in Reagents and Tools Table. *Escherichia coli* strains were cultivated in Luria-Bertani (LB) medium. *M. smegmatis* mc$^2$ 155 strains were cultivated in 7H9 medium (BD Difco, USA) supplemented with 0.2% glycerol and 0.05% Tween 80 (Sigma-Aldrich, Germany), or on 7H10 medium (BD Difco, USA) supplemented with 0.5% glycerol. The culture of *M. tuberculosis* H37Ra strains needs additional 10% OADC. All strains previously mentioned were incubated at 37 °C, and maintained in liquid medium with shaking at 160 rpm. Corresponding antibiotics were added when necessary. The mycobacteriophages used in this study were isolated from soil samples in China by our lab, and the wild-type *M. smegmatis* mc$^2$ 155 strain was used for their propagation. All experiments related to *M. tuberculosis* H37Ra are carried out in a BSL-II laboratory.

## Construction of recombinant mycobacterial strains

pLJR962 (Cas9) vector (Addgene, USA) was used to construct counter-select plasmids as described previously with several

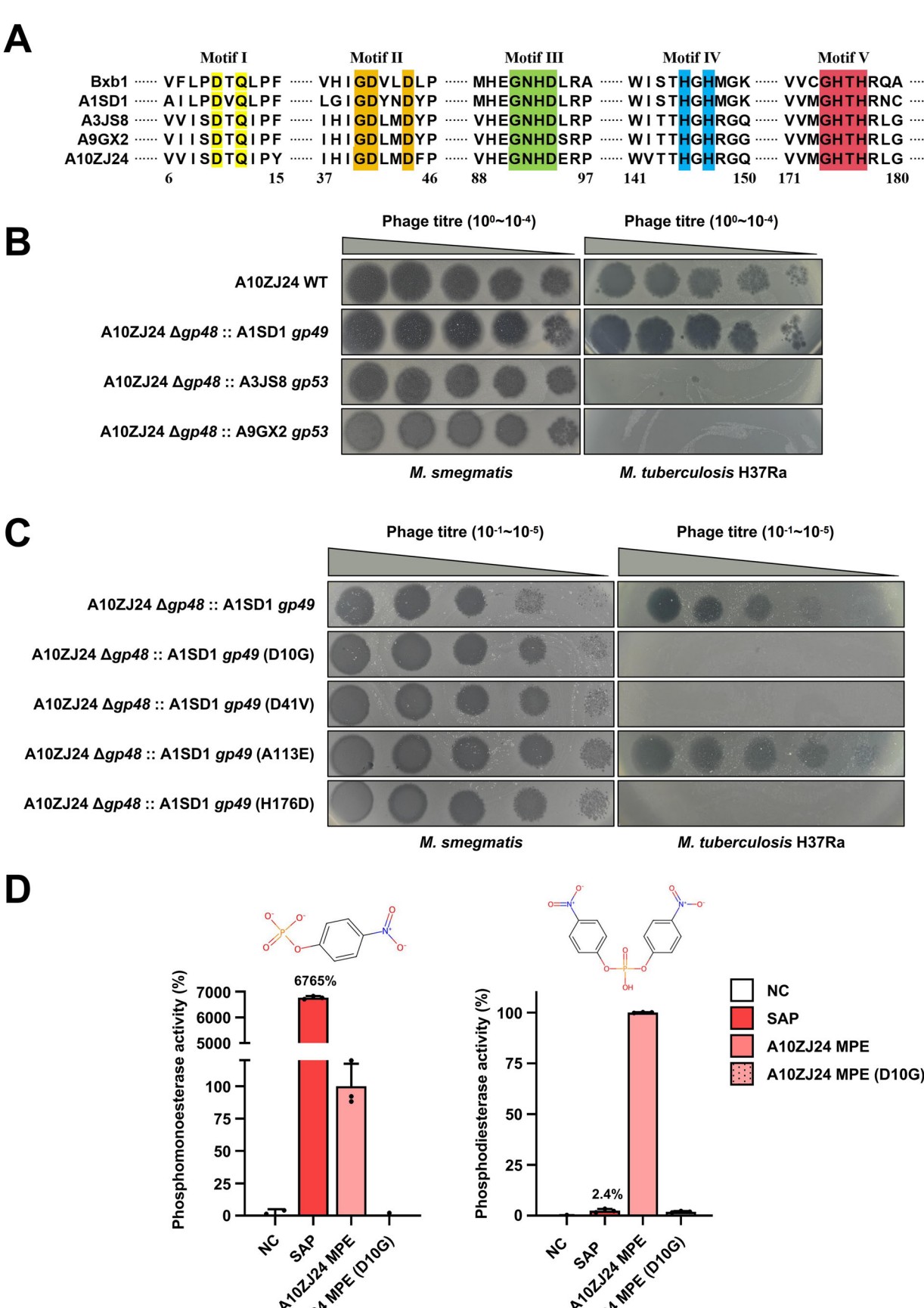

◄   **Figure 4.   Gp48 functions as a conserved MPE-like protein.**

(A) Amino acid sequence alignment of A10ZJ24 Gp48 and other MPE proteins in several Cluster A phages, including Bxb1, A1SD1, A3JS8, A9GX2, and A10ZJ24. Five conserved motifs in the protein are marked with differently colored highlighting. (B) Comparative assays of the growth of recombinant phages on the lawns of *M. tuberculosis* H37Ra when the *gp48* gene in the genome of A10ZJ24 was replaced by a homologous gene of three Cluster A phages: A1SD1 *gp49*, A3JS8 *gp53*, and A9GX2 *gp53*. (C) Effect of different mutations in A1SD1 *gp49* on the growth of the recombinant phage A10ZJ24 on the lawns of *M. tuberculosis* H37Ra. A10ZJ24 Δ*gp48*:: A1SD1 *gp49* represents a positive control in which the A10ZJ24 *gp48* is replaced by A1SD1 *gp49*. (D) Phosphoesterase activity assays of A10ZJ24 Gp48, including phosphomonoesterase (left panel) and phosphodiesterase (right panel) activities. NC indicates the negative control. Alkaline phosphatase (shrimp) was used as positive controls. The activity of A10ZJ24 Gp48 was set at 100%. All data are presented as the mean ± SD ($n = 3$, biological replicates). Source data are available online for this figure.

modifications (Wetzel et al, 2021). Briefly, the plasmid contains a *Streptococcus thermophilus cas9* gene and a small-guide RNA (sgRNA) which can bind target gene. The sgRNA consists of a 20-bp region near the 5′-end of the non-template strand before the corresponding protospacer-adjacent motif. The sgRNA primers were annealed and inserted into the *BsmB* I site of the pLJR962 (Cas9) vector. Subsequently, the recombinant plasmids were separately transferred into *M. smegmatis* mc$^2$ 155 strain and plated on 7H10 plates containing 0.025 mg/mL kanamycin.

To express a phage gene encoding the MPE or its mutants in *M. smegmatis* mc$^2$ 155 and *M. tuberculosis* H37Ra strains, an ATc-inducible expression vector pJR962 was used (Li et al, 2023). Amplified phage genes were separately fused into the modified pJR962 vector between the *Not* I and *Cla* I restriction sites to obtain recombinant plasmids. Next, the recombinant vector were transformed into *M. smegmatis* mc$^2$ 155 or *M. tuberculosis* H37Ra strains, and plated on 7H10 medium containing 0.025 mg/mL kanamycin. The expression vectors containing *M. tuberculosis* H37Ra anti-phage genes also were constructed as mentioned above. The recombinant strains were induced with 0.01 mg/mL ATc for gene complement experiments, and induced with 0.2 mg/mL ATc for gene overexpression.

## Determination of mycobacterial growth curves

To examine the effect of A10ZJ24 Gp48 and its mutant expression on the growth of *M. smegmatis* mc$^2$ 155 or *M. tuberculosis* H37Ra, the growth curves of recombinant strains were determined as described previously with several modifications (Zhu et al, 2018). Briefly, the cells of recombinant strains (OD$_{600}$ = 1.0) were centrifuged and re-suspended in 7H9 medium with 10% OADC and 0.1 mg/mL anhydrotetracycline hydrochloride (ATc) up to an OD$_{600}$ of 0.2, cultivated with 160 rpm at 37 °C. The OD$_{600}$ of *M. smegmatis* mc$^2$ 155 strain's cultures were determined every 4 h, and the OD$_{600}$ of *M. tuberculosis* H37Ra strain's cultures were determined every day.

## Expression and purification of metallophosphoesterase (MPE)

To determine the phosphoesterase activity of MPE, the protein A10ZJ24 Gp48 and its mutant Gp48 (D10G) were expressed and purified. The gene encoding A10ZJ24 Gp48 was amplified by PCR using their specific primers listed in Reagents and Tools Table. The mutant gene encoding A10ZJ24 Gp48 (D10G) was obtained through assembly two fragments containing mutant site by Gibson assembly. Next, the amplified products were separately assembled

into pET28a-SUMO vector and then transformed into ArcticExpress (DE3) pRARE2 (WEIDI, China). The recombinant strains were cultured in LB medium containing 0.025 mg/mL kanamycin, 0.04 mg/mL gentamycin and 0.034 mg/mL chloramphenicol at 37 °C to an OD$_{600}$ of 0.6, and induced with 0.5 mM isopropyl-b-d-thiogalactopyranoside (IPTG) at 16 °C for 16 h. Cells were harvested and proteins were purified as previously described (Li et al, 2023). The eluted proteins were dialyzed at 4 °C for 8 h in buffer containing 20 mM Tris-HCl (pH 8.5), 50 mM NaCl, and 20% glycerol, then stored at −80 °C. Their concentration were determined using the Coomassie Brilliant Blue assay.

## Construction of recombinant phages

BRED strategy was used to construct various phage A10ZJ24 mutants as described previously with several modifications (Marinelli et al, 2008). Briefly, the recombineering substrate DNA fragments (1600–2000 bp), containing both upstream and down-stream sequence of a target gene, were constructed by Gibson assembly. The pJV53 vector with RecE-like and RecT-like gene was transformed into the lysogenic *M. smegmatis* mc$^2$ 155, and its competent cells were prepared according to previously procedures (Li et al, 2023). The mycobacterial cells were electroporated with the above recombineering substrates, recovered at 37 °C for 1–2 h, and plated on top agar lawn of the counter-select strains. Plaques were picked into MP buffer, and were screened by PCR, until successful isolation of the recombinant phages.

## Assays for phage infection ability

The infection efficiencies for mycobacterial strains with phage were evaluated by using a previously reported plaque assay procedure with several modifications (Fullner and Hatfull, 1997). Phages with different dilution gradients, were spotted on the lawns of *M. smegmatis* mc$^2$ 155 cells (OD$_{600}$ = 1.0), and the plaque formation was assessed after incubation at 37 °C overnight. The lawns of *M. bovis* BCG and *M. tuberculosis* H37Ra strains were made by the addition of 1 mL concentrated cultures (OD$_{600}$ = 1.0, concentrate at 10:1). The plaque formation on the lawns of BCG strain was assessed after incubation at 37 °C for 7 days, and on the lawns of *M. tuberculosis* H37Ra for 2–4 days.

## CRISPR/dCas9 interference (CRISPRi) assays

CRISPRi is used to silence the expression of *M. tuberculosis* H37Ra anti-phage genes as described previously with several modifications (Li et al, 2022). Briefly, the CRISPRi plasmids of *Mra_1649*,

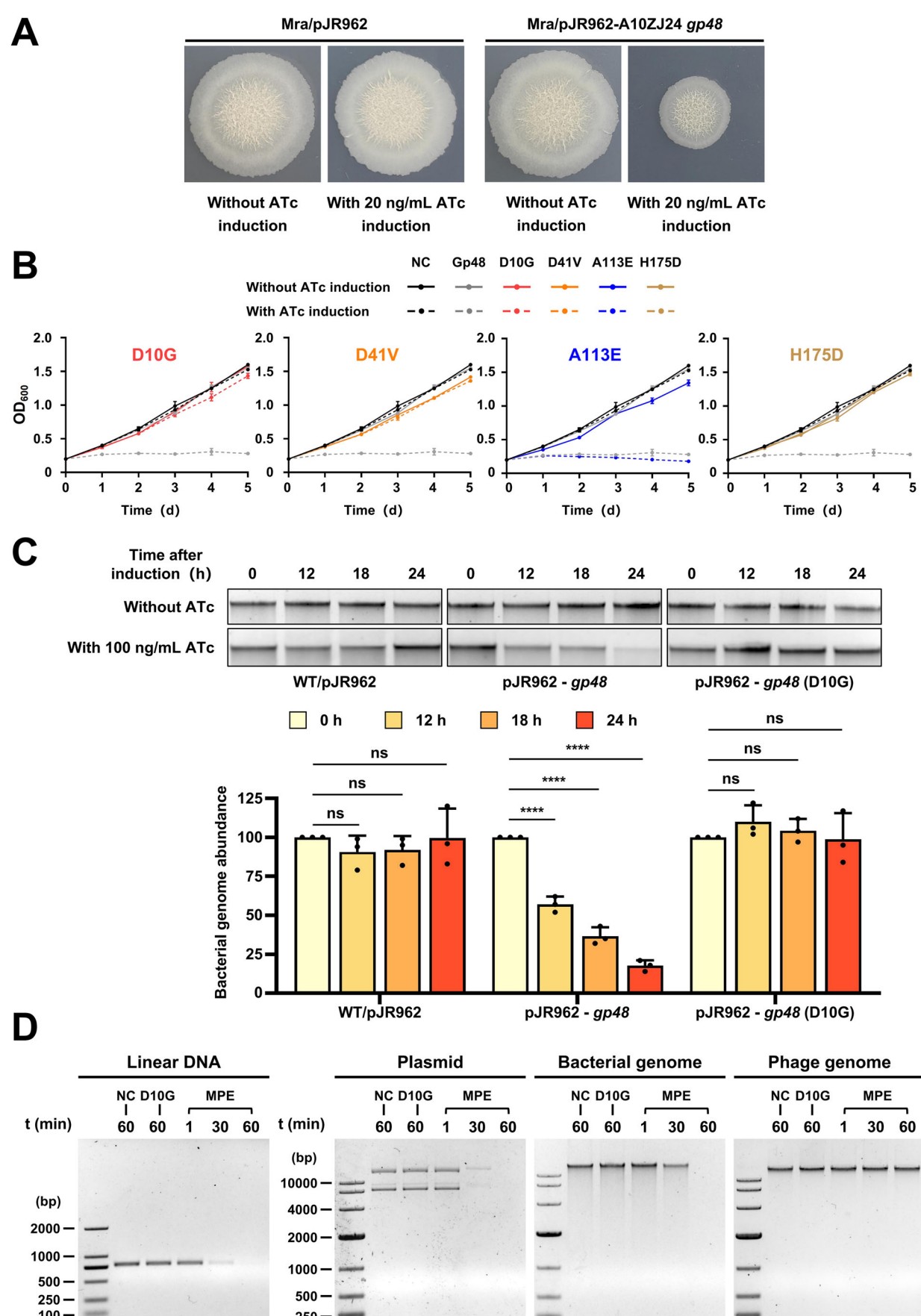

◄ Figure 5. Effect of Gp48 expression on the growth and genomic integrity of *M. tuberculosis* H37Ra.

(A) Representative colony morphologies of the *M. tuberculosis* strain containing empty plasmid pJR962 and recombinant plasmid pJR962-A10ZJ24 *gp48* on 7H10 agar plates, with or without induction with 0.02 mg/mL ATc. (B) Growth curves of recombinant *M. tuberculosis* H37Ra expressing A10ZJ24 Gp48 or its mutant protein. *M. tuberculosis* H37Ra strains harboring the inducible expression vector were grown with or without induction with 0.1 mg/mL ATc. NC represents the wild-type strain containing the empty vector pJR962 used as a negative control. Gp48, D10G, D41V, A113E, and H175D represent strains containing pJR962-*gp48*, pJR962-*gp48* (D10G), pJR962-*gp48* (D41G), pJR962-*gp48* (A113E), and pJR962-*gp48* (H175D) expression vectors, respectively. All data are presented as the mean ± SD ($n = 3$, biological replicates). (C) Assays of the genomic DNA integrity in vivo in *M. tuberculosis* upon ATc-induced expression of A10ZJ24 Gp48 or Gp48 (D10G). Total DNA samples (200 ng) were electrophoresed (upper panel). The relative abundance of complete genomic DNA was calculated according to the concentration of the electrophoresis band determined using Image Lab software (lower panel). All data are presented as the mean ± SD ($n = 3$, biological replicates). The *P* values were calculated by two-way ANOVA using GraphPad Prism v7.0. Asterisks denote significant differences (ns = non-significant, ****$P < 0.0001$) between two groups. Genome abundance of WT/pJR962 at 12 h: $p = 0.4885$, at 18 h: $p = 0.6026$, at 24 h: $p > 0.9999$; Genome abundance of pJR962-*gp48* at 12 h: $p < 0.0001$, at 18 h: $p < 0.0001$, at 24 h: $p < 0.0001$; Genome abundance of pJR962-*gp48* (D10G) at 12 h: $p = 0.4351$, at 18 h: $p = 0.8963$, at 24 h: $p = 0.9962$. (D) In vitro nuclease activity assays of wild-type A10ZJ24 MPE and its mutant proteins. Linear DNA obtained by PCR, plasmids, genomic DNA extracted from bacteria, and the isolated phage A10ZJ24 genome were used as substrates. NC indicates the negative control. The samples were subjected to agarose gel electrophoresis 1, 30, and 60 min after protein incubation with dsDNA. Source data are available online for this figure.

*Mra1940A*, *Mra2329A*, and *Mra_3122* were cloned using pLJR965 vector (Addgene). The plasmid backbone was digested with *BsmBI* and gel purified. sgRNAs were designed to target the non-template strand of the target gene ORF (Reagents and Tools Table). For each individual sgRNA, two complementary oligonucleotides with appropriate sticky end overhangs were annealed and ligated (T4 ligase, Takara) into the *BsmBI*-digested plasmid backbone. These recombinant plasmids were transformed into the wild-type *M. tuberculosis* strain. The recombinant strains were cultured in 7H9 medium supplemented with 0.025 mg/mL kanamycin and 10% OADC until the $OD_{600} = 1.0$, and used for the determination of phage resistance. When the knockdown of genes is necessary, a final concentration of 0.1 mg/mL ATc is added to the culture to induce the expression of sgRNA and dCas9.

## Phage adsorption ability assays

Phage adsorption assays were preformed according to previously reported procedures with several modifications (Fullner and Hatfull, 1997; Barsom and Hatfull, 1996). Briefly, *M. smegmatis* mc$^2$ 155 strains and *M. tuberculosis* H37Ra strains were grown up to an $OD_{600}$ of 1.0 in 7H9 medium supplemented with 10% OADC. The cells centrifuged and re-suspended in 7H9 medium without Tween 80 for three times, and the phage filtrate were added to the cultures at a MOI of 0.1. The mixtures were incubated at 37 °C, and were taken out 1 mL every half an hour. Samples were passed by 0.22 μm filter, and used the diluted samples to determine phage titer by double layer plaque assay. Three sets of sample repeats were included in this assay.

## Assays for the genomic DNA replication of phage

Total DNA extraction of the *M. smegmatis* mc$^2$ 155 strains or *M. tuberculosis* H37Ra strains infected with phages was used to quantify the level of phage replication as described previously with several modifications (Li et al, 2023). Briefly, *M. smegmatis* mc$^2$ 155 strain and *M. tuberculosis* H37Ra strain were infected with wild-type A10ZJ24 or its mutant phage at a MOI of 0.1. The cells were washed three times with 7H9 medium to remove un-adsorbed phage after incubation for 30 min at 37 °C. Samples were taken out every 40 min for *M. smegmatis* mc$^2$ 155 and 6 h for *M. tuberculosis* H37Ra after adsorption. Total DNA was extracted using a genomic

DNA extraction kit (Aidlab) and analyzed by quantitative real-time PCR. The level of phage replication after infection is represented by the level of the intracellular phage circular genome. Primers targeting the regions upstream and downstream of the phage genomic 3′-sticky overhang (CGGCCGGTTA) were designed for PCR amplification.

## Bacterial genome integrity assays

The total DNA of recombinant mycobacterial strains were extracted for determining the effects of A10ZJ24 Gp48 and its mutant proteins on the genomic integrity of *M. smegmatis* mc$^2$ 155 strains and *M. tuberculosis* H37Ra strains according to a previously reported procedure with several modifications (Mayo-Munoz et al, 2022). In brief, the recombinant strains were cultured in 7H9 containing 10% OADC with 160 rpm at 37 °C up to an $OD_{600}$ of 0.8, then the cells were induced with 0.1 mg/mL ATc. Three samples of *M. smegmatis* mc$^2$ 155 strains were collected every 30 min after 2 h induction, and samples of *M. tuberculosis* H37Ra strains were collected out every 6 h after 12 h induction of ATc. Subsequently, Total bacterial DNA was extracted using a genomic DNA extraction kit (Aidlab), and quantified by Nanodrop (Thermo). A total of 200 ng of extracted genomic DNA was loaded onto an agarose gel and electrophoresed in 1×TAE buffer at 100 V for 30 min. Gel images were recorded using ChemiDoc™ XRS+ (Bio-Rad), and the band intensity was determined by image Lab software.

## Phosphoesterase activity assays

The phosphoesterase activity of phage A10ZJ24 Gp48 and its mutant proteins were determined by using 4-Nitrophenyl phosphate (pNPP) and Bis(4-nitrophenyl) phosphate (bis-pNPP) as described previously with several modifications (Dutta et al, 2014). Briefly, the reaction mixtures (200 μl) containing buffer [20 mM Tris-HCl (pH 8.0), 50 mM NaCl, 1 mM MnCl$_2$, 1 mM DTT], 5 mM pNPP (or bis-pNPP), and 0.5 μg of Gp48 or its mutant proteins were incubated for 60 min at 37 °C. Alkaline Phosphatase (Shrimp) was used as a positive control, and dialysis against buffer used as a negative control. Release of para-nitrophenol (pNP) was spectrophotometrically measured at 405 nm, and estimated using its molar extinction coefficient.

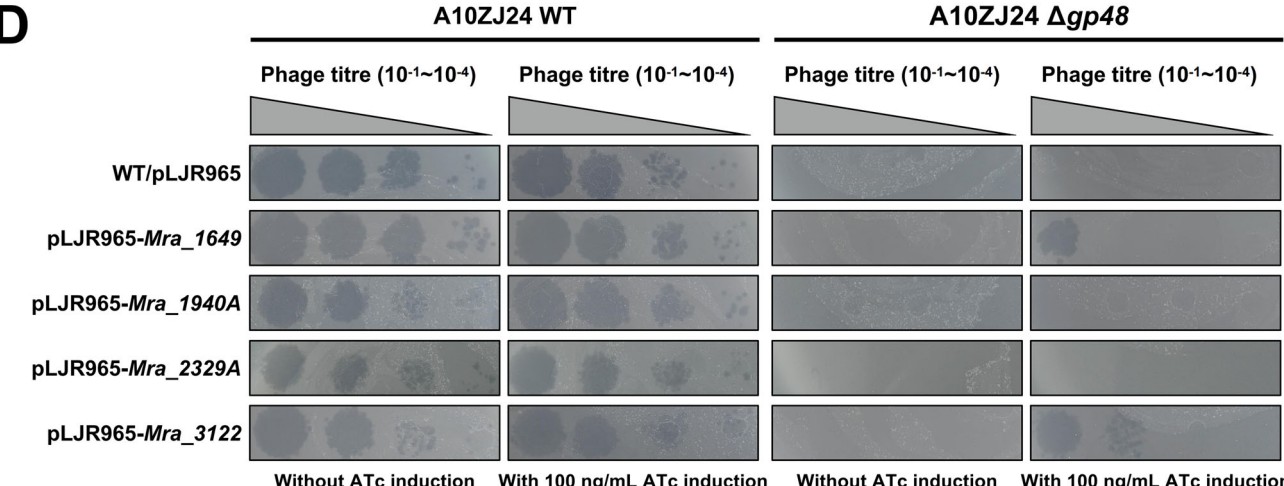

**Figure 6. A10ZJ24 Δ*gp48* infection activates the expression of anti-phage genes of *M. tuberculosis* H37Ra.**

(A) Volcano plot of the differentially expressed genes of *M. tuberculosis* H37Ra triggered by A10ZJ24 Δ*gp48* infection, as determined using RNA sequencing assays. The Cuffdiff program was executed to test for differential expression using the edgR package. The differential expression of a transcript is reported as significant if the test indicated that the false discovery rate–adjusted *P*-value for multiple-testing represented statistically significant values (*q*-value < 0.05). Genes with $\log_2$ (fold-change) values greater than 0.5 were identified as significantly differentially expressed genes. *n* = 3, *n* represents biological replicates. (B) Heat map of the upregulated genes of *M. tuberculosis* H37Ra triggered by infection with A10ZJ24WT and its *gp48*-deletion mutant, for which homologs do not exist in the genome of *M. smegmatis*. WT 1H_1, WT 1H_2, WT 1H_3, Δ*gp48* 1H_1, Δ*gp48* 1H_2, and Δ*gp48* 1H_3 represent three biological replicates 1 h after infection with the phage. WT 10H_1, WT 10H_2, WT 10H_3, Δ*gp48* 10H_1, Δ*gp48* 10H_2, and Δ*gp48* 10H_3, represent three replicates 10 h after infection with the phage. (C) Effect of the expression of several anti-phage genes on the plaque-formation ability of phage A10ZJ24 on the lawns of recombinant *M. tuberculosis* H37Ra. WT/pJR962 represents the wild-type strain containing empty vector pJR962 used as negative control. pJR962-Mra IS6110 represents the ATc-induced strain for *Mra_3420* gene expression. pJR962-Mra Csm represents the strain with co-expression of *Mra_2840*, *Mra_2841*, and *Mra_2842* genes. (D) CRISPRi assays for the effects of the inhibition of the expression of several anti-phage genes on the plaque-forming efficiency of A10ZJ24. WT/pLJR965 represents the wild-type strain containing empty vector pLJR965 used as a negative control. The concentration of ATc used for induction was 0.1 mg/mL. Source data are available online for this figure.

## In vitro nuclease activity assays

Linear DNA, circular plasmids, mycobacterial genomic DNA and A10ZJ24 genomic DNA were used for in vitro nuclease activity assays for A10ZJ24 MPE, as described previously with several modifications (Loeff et al, 2025). Briefly, Linear DNA was amplified by PCR using specific primers targeting A10ZJ24 *gp48*; circular plasmids and mycobacterial genomic DNA were extracted using a DNA rapid extraction kit (Aidlab); and A10ZJ24 genomic DNA was isolated via the standard phenol-chloroform extraction protocol. For the cleavage assays, 200 ng of Linear DNA, circular plasmids, mycobacterial genomic DNA or A10ZJ24 genomic DNA were, respectively, mixed with 0.1 nM of A10ZJ24 MPE in a buffer containing 20 mM Tris-HCl (PH = 8.5), 50 mM NaCl, 1 mM $MnCl_2$ and 1 mM DTT and incubated for the indicated times at 37 °C. After the incubation, the reaction was quenched by the addition of 10× loading buffer (Takara) containing 0.9% SDS. Next, the samples were loaded onto a 1% agarose gel, and were run for 40 min at 100 V, followed by imaging with ChemiDoc™ XRS+ Imaging System (Bio-Rad).

## Transcriptomic analysis

*M. tuberculosis* H37Ra recombinant strains were cultured in 7H9 medium containing 10% OADC up to an $OD_{600}$ of 1.0 with 160 rpm at 37 °C. The cells were infected by wild-type A10ZJ24 or *gp48*-deleted mutant phage at a MOI of 0.1. The samples were harvested after 10 h infection, each strain in three biological replicates. Cells were washed three times in 7H9 medium to remove free phages, and transcriptomic analysis was performed as described previously (Li et al, 2020). In brief, total RNA was isolated using RNA prep Pure Cell/Bacteria kit (Tiangen, China). Strand-specific libraries were prepared using the NEB Next Ultra RNA Library Prep kit for Illumina (Illumina, USA) according to the manufacturer's instructions. Library construction and sequencing were performed at Beijing Novogene Corporation. The statistical significance (*P* value) and fold change of the gene expression between the two groups were calculated using a univariate analysis (*t*-test).

## Western blot assays

To quantify the expression level of A10ZJ24 MPE protein after induced with different concentration of ATc, a series of ATc-inducible fusion expression vectors were constructed. For MPE detection, the FLAG tag was fused to the C-terminus of MPE by inserting its coding sequence immediately upstream of the *gp48* stop codon. The modified MPE-FLAG protein was detected using a mouse monoclonal antibody against the FLAG tag (Sangon Biotech Co., Shanghai, China). And KatG was used as an internal control, as its protein expression level typically remains unchanged (Li et al, 2023). In brief, the recombinant strains were cultured up to an $OD_{600}$ of 0.8, then different concentration of ATc were added. *M. smegmatis* cells were harvested and lysed after 4 h of induction, and *M. tuberculosis* cells were harvested and lysed after 12 h of induction. For western blot assays, the amount of total protein loaded in each lane was the same, and ChemiDoc™ XRS+ (BIO-RAD) was used to quantify the results.

## Quantitative real-time PCR assays

To quantify the gene expression of *M. tuberculosis* H37Ra and A10ZJ24 during phage infection, the mRNAs of *M. tuberculosis* H37Ra strains infected with phage were extracted and qRT-PCR assays were performed as previously described (Barman et al, 2022). The wild-type *M. tuberculosis* strain was cultured in 7H9 medium containing 10% OADC up to an $OD_{600}$ of 1.0. The cells were centrifuged and re-suspended in 7H9 medium (without Tween 80). Subsequently, phage filtrate was added at a MOI of 0.1. The mixtures were incubated at 160 rpm and 37 °C during the infection period. Cells were collected every 4 h post-infection, and total RNA was extracted, reverse-transcribed into cDNA, the analyzed by qRT-PCR. Gene expression levels were normalized to the levels of *16S rRNA* transcripts.

## Biosafety

All experiments were conducted in compliance with the Biosafety Law of China and WHO guidelines. The study utilized *M. tuberculosis* H37Ra (Risk Group 2) in a BSL-2 facility (Certification No. BSL-2021-00143) in Guangxi University, and was approved by China government. The culture of *M. tuberculosis* H37Ra and the disposal of waste indicated a low-level risks. All researchers wore disposable gloves, respirators, lab coats, and completed biosafety training. Waste was autoclaved at 121 °C for 30 min before disposal.

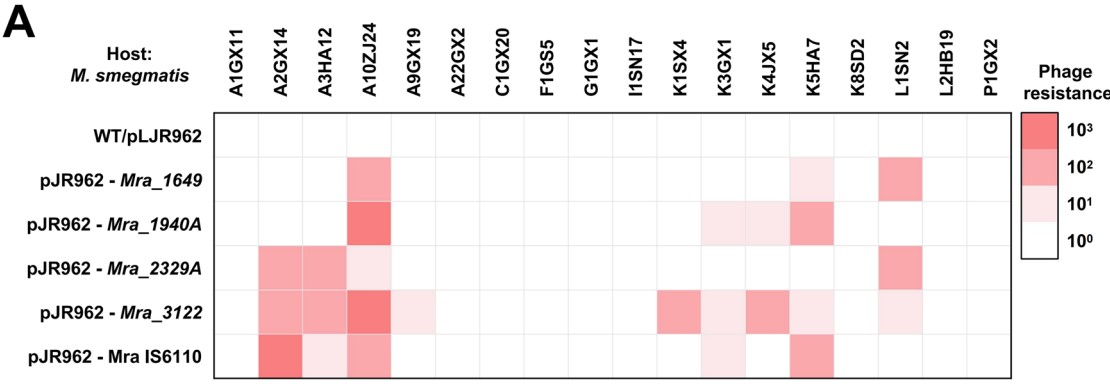

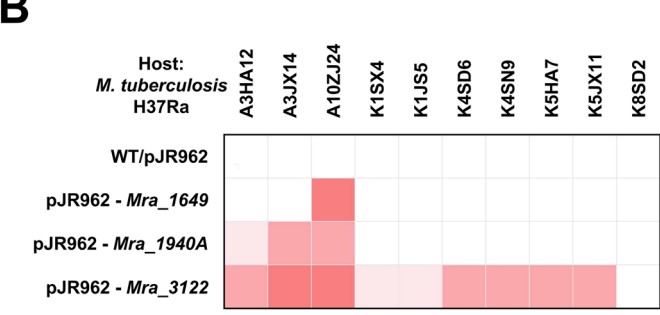

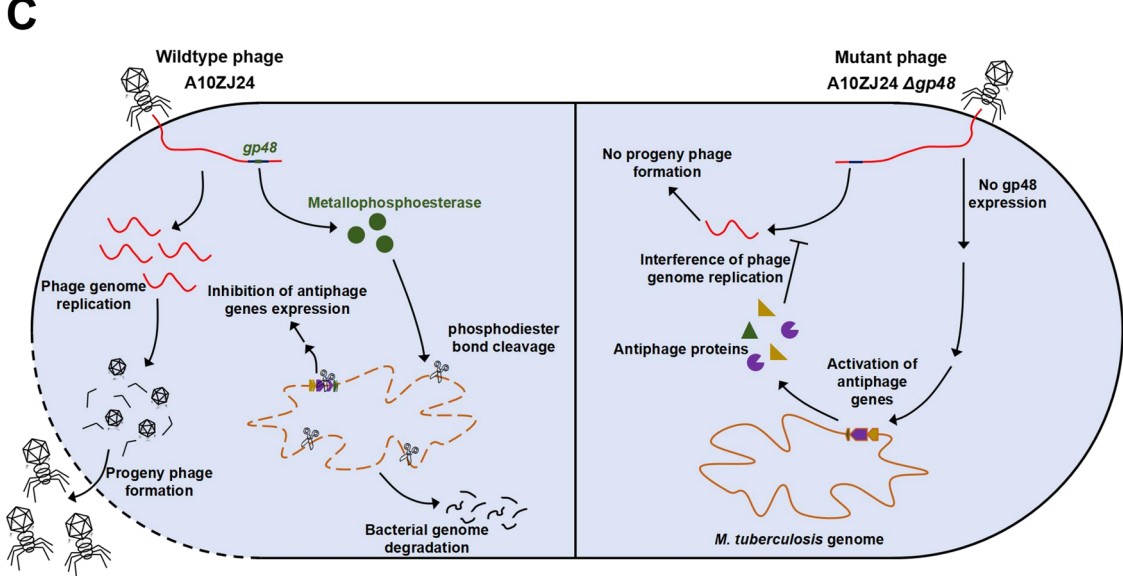

**Figure 7. Defense ability assays for several anti-phage genes identified from *M. tuberculosis*.**

(**A**) Assays for the defense ability of these genes if expressed in *M. smegmatis* mc² 155 strain. The phage infection activity was measured using serial dilution plaque assays and the data are presented in Appendix Fig. S7A. Both pJR962-Mra IS6110 and pJR962-Mra Csm strains were the same as those described in Fig. 6. (**B**) Left, defense ability assays for three genes expressed in *M. tuberculosis*. Serial dilution plaque assay data are shown in Appendix Fig. S7B. Right, effect of *Mra_3122* expression on the plaque-formation ability of phage A10ZJ24 on the lawns of *M. tuberculosis* H37Ra. (**C**) Summary of the anti-phage mechanism triggered by A10ZJ24 *gp48* deletion in *M. tuberculosis*. When the mutant phage A10ZJ24 Δ*gp48* infects *M. tuberculosis*, the *gp48* gene is not expressed and the bacterial genomic DNA integrity is maintained; therefore, multiple anti-phage genes of *M. tuberculosis* can be normally activated by viral infection. This finally inhibited viral genome replication and the progeny phages could not be produced. Source data are available online for this figure.

## Data availability

All data used for this study are publicly available, genome data is deposited in National Center for Biotechnology Information (NCBI) Genbank Database (https://www.ncbi.nlm.nih.gov/Genbank/), with accession numbers PQ130131, PQ130132, PQ130133, and PQ130134), and RNA-seq data is deposited in NCBI's Gene Expression Omnibus and are accessible through GEO Series accession number GSE279196 (https://www.ncbi.nlm.nih.gov/geo/).

The source data of this paper are collected in the following database record: biostudies:S-SCDT-10_1038-S44319-025-00488-4.

## Peer review information

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

## Acknowledgements

This work was supported by the National Key R&D Program of China (2020YFA0907200) and National Natural Science Foundation of China (32230002).

## Author contributions

**Junlong Li**: Conceptualization; Resources; Data curation; Formal analysis; Supervision; Validation; Investigation; Visualization; Methodology; Writing—original draft; Project administration; Writing—review and editing. **Yihao Song**: Formal analysis; Investigation; Visualization; Methodology. **Xiao Guo**: Resources; Formal analysis; Visualization; Methodology. **Zheng-Guo He**: Conceptualization; Resources; Data curation; Formal analysis; Supervision; Funding acquisition; Validation; Investigation; Methodology; Writing—original draft; Project administration; Writing—review and editing.

Source data underlying figure panels in this paper may have individual authorship assigned. Where available, figure panel/source data authorship is listed in the following database record: biostudies:S-SCDT-10_1038-S44319-025-00488-4.

## Disclosure and competing interests statement

The authors declare no competing interests.

