## [Peer Review File · EMBO Reports]

A conserved phage phosphoesterase enables evasion of bacterial antiviral immunity

Junlong Li, Yihao Song, Xiao Guo, and Zheng-Guo He

Corresponding author(s): Zheng-Guo He (hezhengguo2024@whu.edu.cn)

Review Timeline:

Submission Date:	18th Nov 24
Editorial Decision:	17th Jan 25
Revision Received:	24th Mar 25
Editorial Decision:	5th May 25
Revision Received:	15th May 25
Accepted:	16th May 25

Editor: Achim Breiling

Transaction Report:

Dear Prof. He,

Thank you for the transfer of your manuscript to EMBO reports. I have now received the reports from the three referees that were asked to evaluate your study, which can be found at the end of this email.

As you will see, the referees think that these findings are of interest. However, they have several comments, concerns, and suggestions, indicating that a major revision of the manuscript is necessary to allow publication of the study in EMBO reports. As the reports are below, and all the referee concerns need to be addressed, I will not detail them here. However, as indicated by the referees, the text requires thorough correction and rewriting by a native speaker who is also familiar with the topic. Please have this done.

Given the constructive referee comments, I would like to invite you to revise your manuscript with the understanding that the concerns of the referees must be addressed in the revised manuscript and in a detailed point-by-point response. Acceptance of your manuscript will depend on a positive outcome of a second round of review. It is EMBO reports policy to allow a single round of revision only and acceptance of the manuscript will therefore depend on the completeness of your responses included in the next, final version of the manuscript.

- 1) a .docx formatted version of the final manuscript text (including legends for main figures, EV figures and tables), but without the figures included. Figure legends should be compiled at the end of the manuscript text.
- 2) individual production quality figure files as .eps, .tif, .jpg (one file per figure), of main figures and EV figures. Please upload these as separate, individual files upon re-submission.

- 4) a complete author checklist, which you can download from our author guidelines

(<https://www.embopress.org/page/journal/14693178/authorguide>). Please insert page numbers in the checklist to indicate where the requested information can be found in the manuscript. The completed author checklist will also be part of the RPF.

5) that primary datasets produced in this study (e.g. RNA-seq, CHIP-seq, structural and array data) are deposited in an appropriate public database. If no primary datasets have been deposited, please also state this in a dedicated section (e.g. 'No primary datasets have been generated and deposited'), see below.

The accession numbers and database should be listed in a formal "Data Availability" section that follows the model below. This is now mandatory (like the COI statement). Please note that the Data Availability Section is restricted to new primary data that are part of this study. This section is mandatory. As indicated above, if no primary datasets have been deposited, please state this in this section

Data availability

8) Regarding data quantification and statistics, please make sure that the number "n" for how many independent experiments were performed, their nature (biological versus technical replicates), the bars and error bars (e.g. SEM, SD) and the test used to calculate p-values is indicated in the respective figure legends (also for EV and Appendix figures). Please also check that all the p-values are explained in the legend, and that these fit to those shown in the figure. Please provide statistical testing where applicable. Please avoid the phrase 'independent experiment', but clearly state if these were biological or technical replicates. Please also indicate (e.g. with n.s.) if testing was performed, but the differences are not significant. In case n=2, please show the data as separate datapoints without error bars and statistics. See also: <http://www.embopress.org/page/journal/14693178/authorguide#statisticalanalysis>

9) Please add scale bars of similar style and thickness to microscopic images, using clearly visible black or white bars (depending on the background). Please place these in the lower right corner of the images themselves. Please do not write on or near the bars in the image but define the size in the respective figure legend.

10) Please also note our reference format:

12) We now use CRediT to specify the contributions of each author in the journal submission system. CRediT replaces the author contribution section. Please use the free text box to provide more detailed descriptions and do NOT provide your final manuscript text file with an author contributions section. See also our guide to authors: <https://www.embopress.org/page/journal/14693178/authorguide#authorshipguidelines>

13) All Materials and Methods need to be described in the main text using our 'Structured Methods' format, which is required for all research articles. According to this format, the Methods section should include a Reagents and Tools Table (listing key reagents, experimental models, software, and relevant equipment and including their sources and relevant identifiers), uploaded as separate file, and a Methods section in which we encourage the authors to describe their methods using a step-by-step protocol format with bullet points, to facilitate the adoption of the methodologies across labs. More information on how to adhere to this format as well as downloadable templates (.doc) for the Reagents and Tools Table can be found in our author guidelines (section 'Structured Methods'):

14) Please order the sections like this, using these names:

Title page - Abstract - Keywords - Introduction - Results & Discussion - Methods - Data availability section - Acknowledgements (including funding information) - Disclosure and Competing Interests Statement - References - Figure legends - Expanded View Figure legends

15) Please make sure that all the funding information is also entered into the online submission system and that it is complete and similar to the one in the acknowledgement section of the manuscript text file.

I look forward to seeing a revised form of your manuscript when it is ready.

Yours sincerely,

Referee #1:

The manuscript of Li et al on "A conserved phage phosphoesterase enables evasion of host bacterial antiviral immunity" describes the identification of non-essential genes of a temperate mycobacteriophage A10ZJ24. A modified BRED method was first utilized to produce a series mutant phages with single gene-deletions. All these mutants reproduced efficiently in *M. smegmatis*, however, some showed clear growth defects in *M. bovis* and *M. tuberculosis*, and the one with gene *g48* deletion failed to reproduce in *M. tuberculosis*. With well-designed experiments the authors demonstrated that Gp48 is required for phage genome replication in *M. tuberculosis*. Bioinformatics identified Gp48 as a metallophosphoesterase and that activity was indirectly confirmed by generating selected active-site single-amino acid substitution mutants. Then Gp48 was shown to be responsible of host strain genomic DNA degradation, and thereby inactivate unknown host anti-phage defense systems that were identified by RNA-seq analysis.

Overall, the manuscript presents new information on phage strategy to evade host anti-phage defense.

While the scientific contents of the manuscript are worth publication, the scientific presentation, the usage of English, does not reach the required level, and this is the case throughout the manuscript. The text requires a thorough overhaul by a native English speaker who is also familiar with the topic in question.

Referee #2:

This is an interesting work on the mycobacteriophage A10ZJ24 where Li et al have used a modified BRED strategy to knockout phage genes when integrated as a lysogen. They were able to successfully make deletions in 42 non-essential genes. These

mutant phages were then tested for their ability to form plaques on *M. smegmatis*, *M. bovis* and *Mtb H37Ra*. Deletion of gp48 resulted in impairment in the ability of the mutant to form viable plaques on *M. bovis* and *Mtb H37Ra*, but not on *M. smegmatis* suggesting that gp48 is required for A10ZJ24 to infect *M. bovis* and *Mtb H37Ra*. The gp48 mutant phage was found to retain its ability to adsorb and infect *Mtb* but was impaired in the subsequent step of replication. The authors also showed that gp48 encodes an MPE like protein that degrades host genomic DNA thereby preventing expression of host encoded anti-phage genes.

Major points:

1. This is a thorough piece of work that could greatly benefit from rewriting and better articulation.
2. How rapidly is the host DNA degraded? Its hard to comprehend how phages are still being replicated in a host where genomic DNA is completely degraded. Would this phage be able to infect non replicating cells eg those treated with antibiotic?
3. Would a *Mtb* strain with a mutation in the anti-phage genes recapitulate WT?

Referee #3:

The manuscript by Li et al. is devoted to the study of the functional role of the metallophosphoesterase-like protein (Gp48) from mycobacteriophage A10ZJ24. Through a gene essentiality assay, the authors showed that deletion of this gene doesn't affect the infection of *M. smegmatis* by this phage but cancels its ability to infect *M. bovis* BCG and *M. tuberculosis* H37Ra. The authors explained this effect by the phosphoesterase activity of Gp48 and the degradation of genomic DNA containing anti-phage genes in *M. tuberculosis*.

Despite the interesting observations, there are several gaps between the presented results and the final hypothesis (presented in Figure 7 and in the abstract).

Major Points:

"Gp48 is expressed in the early stage of phage infection and disrupts the mycobacterial chromosomal DNA."

A bacterial DNA integrity assay was performed only for the overexpression of Gp48 under induction, but not after infection with the phages (wild-type and with deletion of gp48). These experiments should be done for different strains and phages to demonstrate the direct effect of Gp48 on DNA degradation after infection.

The reviewer recommends looking at changes in metabolites; maybe there are other phosphorylated targets for Gp48 and not genomic DNA.

The authors should also demonstrate DNA cleavage by A1SD1 Gp48 in vitro.

"The gp48-deleted phage can normally infect and invade *M. tuberculosis*. However, it does not prevent the activation of anti-phage genes, resulting in a loss of its genome DNA replication ability in *M. tuberculosis*."

This conclusion is also not supported by the presented data.

The authors showed higher expression of several genes of *M. tuberculosis* after infection with the phage with deletion of gp48 and also demonstrated their potential role in anti-defense against phages. However, to show that Gp48 "prevents the activation of anti-phage genes," the authors should perform the same experiment for the wild-type phage with gp48 and compare the expression of the host genes.

Referee #1:

The manuscript of Li et al on "A conserved phage phosphoesterase enables evasion of host bacterial antiviral immunity" describes the identification of non-essential genes of a temperate mycobacteriophage A10ZJ24. A modified BRED method was first utilized to produce a series mutant phages with single gene-deletions. All these mutants reproduced efficiently in *M. smegmatis*, however, some showed clear growth defects in *M. bovis* and *M. tuberculosis*, and the one with gene g48 deletion failed to reproduce in *M. tuberculosis*. With well-designed experiments the authors demonstrated that Gp48 is required for phage genome replication in *M. tuberculosis*. Bioinformatics identified Gp48 as a metallophosphoesterase and that activity was indirectly confirmed by generating selected active-site single-amino acid substitution mutants. Then Gp48 was shown to be responsible of host strain genomic DNA degradation, and thereby inactivate unknown host anti-phage defense systems that were identified by RNA-seq analysis.

Overall, the manuscript presents new information on phage strategy to evade host anti-phage defense.

While the scientific contents of the manuscript are worth publication, the scientific presentation, the usage of English, does not reach the required level, and this is the case throughout the manuscript. The text requires a thorough overhaul by a native English speaker who is also familiar with the topic in question.

Response: Thank you for your positive comments and constructive suggestion. In this submission, we have invited a native English speaker who is also familiar with the topic to revise the manuscript (please see the attached file). We believed that both the language and the scientific presentation for the new manuscript are much improved.

Referee #2:

This is an interesting work on the mycobacteriophage A10ZJ24 where Li et al have used a modified BRED strategy to knockout phage genes when integrated as a lysogen. They were able to successfully make deletions in 42 non-essential genes. These mutant phages were then tested for their ability to form plaques on *M. smegmatis*, *M. bovis* and *Mtb H37Ra*. Deletion of gp48 resulted in impairment in the ability of the mutant to form viable plaques on *M. bovis* and *Mtb H37Ra*, but not on *M. smegmatis* suggesting that gp48 is required for A10ZJ24 to infect *M. bovis* and *Mtb H37Ra*. The gp48 mutant phage was found to retain its ability to adsorb and infect *Mtb* but was impaired in the subsequent step of replication. The authors also showed that gp48 encodes an MPE like protein that degrades host genomic DNA thereby preventing expression of host encoded antiphage genes.

Response: Thank you very much for your positive comments.

Major points:

1. This is a thorough piece of work that could greatly benefit from rewriting and better articulation.

Response: Thank you for your constructive comments. In this submission, we have invited a native English speaker to revise and rewrite the manuscript (please see the attached file). We believed that both the language and the scientific presentation for new manuscript are much improved.

2. How rapidly is the host DNA degraded? Its hard to comprehend how phages are still being replicated in a host where genomic DNA is completely degraded. Would this phage be able to infect non replicating cells eg those treated with antibiotic?

Response: Thank you for your comments. As shown in Fig. 5C, after inducing expression of Gp48 in Mtb strain for 12 hours, the bacterial genomic DNA was observed degradation. As shown in Fig. 5D, in vitro enzyme activity assays indicate that Gp48 protein can completely degrade 200ng of bacterial genomic DNA at a concentration of 50pmol for 60 minutes. Based on the results of the expression of Gp48 protein in Mtb after 4 hours of phage infection in Fig. 3B, we speculate that the host's genomic DNA can be completely degraded within 12 hours after phage infection. Actually, previous studies have reported that bacteriophages degrade host genomic DNA in the early stages of infection to assist in their own replication. For example, the genome DNA of P. chloriraphis was observed degradation 20 minutes after infection of its bacteriophage 201-2-1, but at this time the bacteriophage has not yet begun its replication process (Reference 1, 2). In addition, studies have shown that the production of bacteriophage particles can be synthesized directly through cell-free systems without relying on complete bacterial cells and genomes (Reference 3, 4).

1. Mendoza SD et al. A bacteriophage nucleus-like compartment shields DNA from CRISPR nucleases. Nature. 2020 Jan;577(7789):244-248.
2. Erb ML et al. A bacteriophage tubulin harnesses dynamic instability to center DNA in infected cells. Elife. 2014 Nov 27;3:e03197.
3. Silverman AD et al. Cell-free gene expression: an expanded repertoire of applications. Nat Rev Genet. 2020 Mar;21(3):151-170.
4. Shin J et al. Genome replication, synthesis, and assembly of the bacteriophage T7 in a single cell-free reaction. ACS Synth Biol. 2012 Sep 21;1(9):408-13.

3. Would a Mtb strain with a mutation in the antiphage genes recapitulate WT?

Response: Thank you for your comments. In this revision, we have added further experiments to address your concerns. Due to the extremely slow growth of Mycobacterium tuberculosis (Mtb), it takes a very long time to directly mutate or knock out its genes and successfully obtain mutant strains. Alternatively, we have already used CRISPRi technology to suppress several selected resistance genes at the transcriptional level, and found that A10ZJ24 Δ gp48 can indeed restore the infectivity of Mtb strain to varying degrees (Fig5D). This is very consistent with our model.

Referee #3:

The manuscript by Li et al. is devoted to the study of the functional role of the metallophosphoesterase-like protein (Gp48) from mycobacteriophage A10ZJ24. Through a gene essentiality assay, the authors showed that deletion of this gene doesn't affect the infection of *M. smegmatis* by this phage but cancels its ability to infect *M. bovis* BCG and *M. tuberculosis* H37Ra. The authors explained this effect by the phosphoesterase activity of Gp48 and the degradation of genomic DNA containing anti-phage genes in *M. tuberculosis*.

Despite the interesting observations, there are several gaps between the presented results and the final hypothesis (presented in Figure 7 and in the abstract).

Response: Thank you for your comments.

Major Points:

"Gp48 is expressed in the early stage of phage infection and disrupts the mycobacterial chromosomal DNA."

A bacterial DNA integrity assay was performed only for the overexpression of Gp48 under induction, but not after infection with the phages (wild-type and with deletion of gp48). These experiments should be done for different strains and phages to demonstrate the direct effect of Gp48 on DNA degradation after infection.

Response: Thank you for your comments. Sorry, we did not accurately state the conclusion of the article. In this revision, the sentence has been changed to: Gp48 is expressed in the early stage of phage infection and its purified protein can disrupt the mycobacterial chromosomal DNA.

Actually, it is difficult to obtain accurate conclusions by directly infecting bacteria with bacteriophages and measuring genomic degradation. Firstly, it is difficult to ensure that every bacterial cell is infected in an experiment, so the total amount of genomic DNA will change with the number of infected bacteria, making it difficult to compare DNA degradation results with each other; Secondly, previous studies have shown that phage genomes may encode multiple proteins with DNA degradation activity (such as lambda bacteriophages), which will function at different stages.

The reviewer recommends looking at changes in metabolites; maybe there are other phosphorylated targets for Gp48 and not genomic DNA.

Response: Thank you for your suggestion. However, our series of experimental results strongly indicate that Gp48 cleavage of genomic DNA leads to ineffective expression of Mtb encoded anti phage genes, which is the fundamental mechanism that helps bacteriophages escape Mtb defense, and is unlikely to be caused by changes in metabolites. Firstly, based on sequence conservation analysis, Gp48 belongs to the metal phosphatase superfamily (please see Fig4A). Secondly, our series of in vivo and in vitro experimental results have confirmed that Gp48 indeed exhibits strong phosphodiesterase activity (please see Fig4D, Fig5C, and Fig5D). Thirdly, we identified multiple anti phage genes upregulated after A10ZJ24 Δ gp48 infection through extensive comparative transcriptomic analysis and validated their defense against A10ZJ24 (please see Fig6C, FigS6A). Finally, consistent with our model, when further inhibiting the expression of these phage resistant genes separately, the infection ability of A10ZJ24 Δ gp48 against Mtb strain can be restored to varying degrees (please see Fig5D). Therefore, based on these bioinformatics, physiological,

biochemical and genetic data together with omics analysis results, they clearly support our model that the reason why mutant bacteriophages lacking Gp48 lose their ability to infect Mtb is that they cannot effectively destroy the host genome and inhibit the expression of resistance genes.

The authors should also demonstrate DNA cleavage by A1SD1 Gp48 in vitro.

Response: Thank you for your constructive suggestion. We have added in vitro analysis based on your comments. As shown in FigS4B and 5D, we purified A10ZJ24 Gp48 and its mutant proteins and determined their in vitro activity. The results showed that Gp48, rather than the mutant protein, could completely degrade linear DNA, plasmids, and bacterial genomic DNA within 1 hour. These results are completely consistent with our in vivo analysis results mentioned above.

"The gp48-deleted phage can normally infect and invade *M. tuberculosis*. However, it does not prevent the activation of anti-phage genes, resulting in a loss of its genome DNA replication ability in *M. tuberculosis*."

This conclusion is also not supported by the presented data.

The authors showed higher expression of several genes of *M. tuberculosis* after infection with the phage with deletion of gp48 and also demonstrated their potential role in anti-defense against phages. However, to show that Gp48 "prevents the activation of anti-phage genes," the authors should perform the same experiment for the wild-type phage with gp48 and compare the expression of the host genes.

Response: Thank you for your suggestion. In this revision, we have added experimental data based on your comments. We complemented transcriptomic analysis on Mtb strains infected with A10ZJ24 WT and A10ZJ24 Δ gp48 bacteriophages, and updated Fig6B. As shown in the figure, 40 unique genes in Mtb were significantly upregulated after A10ZJ24 Δ gp48 infection, while there was no significant difference or small change after A10ZJ24 WT infection (detailed data can be found in the uploaded GSE279196 file). Furthermore, we conducted qRT PCR validation on the screened and identified resistance genes, including *Mra_1649*, *Mra_1940A*, *Mra_2329A*, and *Mra3122*. As shown in FigS6B, these genes were significantly upregulated after A10ZJ24 Δ gp48 infection, while there was no significant change after A10ZJ24 WT infection. Thanks again.

Dear Prof. He,

Thank you for the submission of your revised manuscript to our editorial offices. I have now received the reports from the three referees that were asked to re-evaluate the study, you will find below. As you will see, the referees now support the publication of your manuscript in EMBO reports. However, referees #1 and #2 have remaining concerns and suggestions to improve the manuscript, I ask you to address in a final revised manuscript. Please provide a final p-b-p-response to the remaining issues and the editorial requests below.

I have these editorial requests:

- I would suggest this shortened title:

A conserved phage phosphoesterase enables evasion of bacterial antiviral immunity

- Please have your final manuscript carefully proofread by a native speaker. See also the points of referee #1.

- Please provide the abstract written on present tense throughout.

- We now use CRediT to specify the contributions of each author in the journal submission system. CRediT replaces the author contribution section. Please use the free text box to provide more detailed descriptions and do NOT provide your final manuscript text file with an author contributions section. See also our guide to authors:

<https://www.embopress.org/page/journal/14693178/authorguide#authorshipguidelines>

- Please check again that the number "n" for how many independent experiments were performed, their nature (biological versus technical replicates), the bars and error bars (e.g. SEM, SD) and the test used to calculate p-values is indicated in the respective figure legends. Please also check that all the p-values are explained in the legend, and that these fit to those shown in the figure. Please provide statistical testing where applicable. Please avoid the phrase 'independent experiment' but clearly state if these were biological or technical replicates. Please also indicate (e.g. with n.s.) if testing was performed, but the differences are not significant. In case n=2, please show the data as separate datapoints without error bars and statistics. See also:

<http://www.embopress.org/page/journal/14693178/authorguide#statisticalanalysis>

If n<5, please show single datapoints for diagrams. It seems presently many diagrams are missing the 'n.s.'. Moreover:

- Please provide the exact p values in the legends of figures 1B, 3B-D; 5C.

- Please name the Methods section just 'Methods' and remove the 'Reagents and Tools Table' from the Methods section of the main manuscript text file. This should only be uploaded separately. Please remove the instructions from the final Reagents and Tools Table.

- Please add the information provided in the EV tables (Table EV1 and EV2) to the reagents and tools table and remove the two files. Please update any callouts and/or provide callouts to the Reagents and Tools Table as appropriate.

- There are callouts to Tables S1-S3 and S4 in the manuscript, but no such tables are provided (in the Appendix). Please check, update the callouts or provide these tables.

- Please remove the section 'Supplementary Material' from the manuscript.

- Please remove now the referee token from The Data Availability section (DAS) and make sure that all datasets are public latest on the date of online publication of the study.

- Please make sure that all the funding information is also entered into the online submission system and that it is complete and similar to the one in the acknowledgement section of the manuscript text file. Presently, the grant from the National Key R&D Program of China (2020YFA0907200) is missing from the submission system. Please check.

- Please add a paragraph to the methods section (titled 'Biosafety') providing details on where and how biosafety-relevant experiments were performed and that these were approved, and by whom (institution, government).

- You indicate in the checklist that you used a select agent (a biological agent or a toxin has the potential to pose a severe threat) in the study (<https://www.selectagents.gov/sat/list.htm>). If this is true, please declare which agents have been used, that security measures have been applied and which ones, and that you have approval for the use (institutional, governmental). If no select agent has been used, please remove that part from the checklist (changing the entry to 'Not Applicable').

- During our figure integrity check, we noted a reuse of the rightmost panel in Fig. 1B (gp48) in Appendix Fig. S1 (A10ZJ24

deltagp48). Please check. If this is intentional, please clearly state this in the respective figure legends.

- We also noted overlapping features between some of the images in 6C, 7B and Appendix Fig. S7, which might be caused by the rather low image quality. The contrast of most of plaque-formation assays shown in Figs. 6, 7 and S7 is rather low. Could this be improved?

- Moreover, there is an image reuse within Appendix Fig. S1. Panel 'A10ZJ24 deltagp53' and 'A10ZJ24 deltagp54' seem identical. Please check.

In addition, I would need from you uploaded separately:

Best,

Referee #1:

The manuscript of Li et al on "A conserved phage phosphoesterase enables evasion of host bacterial antiviral immunity" describes the identification of non-essential genes of a temperate mycobacteriophage A10ZJ24. A modified BRED method was first utilized to produce a series of mutant phages with single gene-deletions. All these mutants reproduced efficiently in *M. smegmatis*, however, some showed clear growth defects in *M. bovis* and *M. tuberculosis*, and the one with gene g48 deletion failed to reproduce in *M. tuberculosis*. With well-designed experiments the authors demonstrated that Gp48 is required for phage genome replication in *M. tuberculosis*. Bioinformatics identified Gp48 as a metallophosphoesterase and that activity was indirectly confirmed by generating selected active-site single-amino acid substitution mutants. Then Gp48 was shown to be responsible of host strain genomic DNA degradation, and thereby inactivate unknown host anti-phage defense systems that were identified by RNA-seq analysis. Overall, the manuscript presents new information on phage strategy to evade host anti-phage defense.

The English usage of the revised manuscript has been improved, however, I still found a few points where the authors need to revise the text:

L36-40. These two sentences are confusing. Perhaps better like this: While the gp48-mutant phage infected and injected normally its DNA into *M. tuberculosis* cells, it was not able to prevent the activation of the bacterial anti-phage genes which inhibited the replication of the mutant phage genomic DNA.

L83. Remove extra left parenthesis sign

L207-213. The text does not make sense and does not reflect the results in figure 4B. And I found out that it is due to an omission of a single word "not" from the sentence: both A9GX2 gp53 and A3JS8 gp53 could NOT replace A10ZJ24 gp48 to produce clear plaques on the lawns of *M. tuberculosis* H37Ra.

L216-7. Revise the text as you produced three replacement mutants.

L223, and see L259. was removed from the conserved motifs > was located outside of the conserved motifs

L246: grow > growth

L247, and elsewhere: is the ATc concentration indeed nanograms/mL, generally it is used as micrograms per mL. This comment is also valid for the antibiotic concentrations mentioned in the Mat & Meth section. L427 forward

L482. incubated > incubation

L486. The last sentence does not make sense

L500. Check the tense of the last sentence

L517. Described previously

L519-20. Revise the sentence

L523-525. The last two sentences need revision

L535-537. The sentence needs revision

L541. Also this sentence

L547. PH > pH

L558. Briefly, linear DNA was ...
L560. genomic DNA was ...
L581. What are meant by metabolites?
L586. vectors
L590-1. The sentence needs revision
L593. At what time were the cells collected?
L603-605. The sentence needs revision
L785. The sentence is unclear
L833. shadows > highlighting
L842. SAP is Shrimp alkaline phosphatase, so why give it twice?
L895. sed > used

Referee #2:

The current revision addresses several of the previous comments and illustrates an interesting interaction between phage genes and antidefense mechanisms which determines host range. A few changes in writing style would make the paper easier to read: In several places the authors have summarized the conclusion of the section in the opening statement. This is confusing to read. eg :Ln 196-213 and 288-300. Additionally, section 288-300 gives the impression that RNAseq of only the gp48 mutant was conducted. It isn't until the last sentence that one realizes that WT RNAseq was also conducted. Rewriting both these sections would improve clarity of the paper. Also rewriting section 301-312 would be beneficial.

Ln 257: Why is gp48 toxic to *M. smegmatis*? This is not consistent with the remaining data.

Ln 278-282: Why is phage DNA not degraded by purified gp48?

Ln 304, Fig 6C : *Mra_2329A* is not seen in figure. Only 1649 and 3122 have an effect. The text is misleading. Similarly Ln 313 and Figure 6D - only 1649 and 3122 have an effect.

Referee #3:

The manuscript suitable for publication in EMBO reports without further revision.

Response to the Editor and Reviewers:

Editor:

- I would suggest this shortened title:

A conserved phage phosphoesterase enables evasion of bacterial antiviral immunity

Response: Thanks. The title has been revised according to the suggestion.

- Please have your final manuscript carefully proofread by a native speaker. See also the points of referee #1.

Response: The manuscript has been proofread as you suggested.

- Please provide the abstract written on present tense throughout.

Response: The tense of the abstract has been modified.

- We now use CRediT to specify the contributions of each author in the journal submission system. CRediT replaces the author contribution section. Please use the free text box to provide more detailed descriptions and do NOT provide your final manuscript text file with an author contributions section. See also our guide to authors: <https://www.embopress.org/page/journal/14693178/authorguide#authorshipguidelines>

Response: In this revision, the author contribution section in manuscript has been removed. Instead, we provides a CRediT file to specify the contributions of each author in the journal submission system.

- Please check again that the number "n" for how many independent experiments were performed, their nature (biological versus technical replicates), the bars and error bars (e.g. SEM, SD) and the test used to calculate p-values is indicated in the respective figure legends. Please also check that all the p-values are explained in the legend, and that these fit to those shown in the figure. Please provide statistical testing where applicable. Please avoid the phrase 'independent experiment' but clearly state if these were biological or technical replicates. Please also indicate (e.g. with n.s.) if testing was performed, but the differences are not significant. In case n=2, please show the data as separate datapoints without error bars and statistics. See also:

<http://www.embopress.org/page/journal/14693178/authorguide#statisticalanalysis>

If n<5, please show single datapoints for diagrams. It seems presently many diagrams are missing the 'n.s.'. Moreover:

Response: In this revision, we have already checked the above parts of figure legends.

- Please provide the exact p values in the legends of figures 1B, 3B-D; 5C.

Response: The *P* values of figures have been provided, but the exact *P* value cannot be provided when $p < 0.0001$ or $p > 0.9999$ (Calculated using GraphPad Prism 7).

- Please name the Methods section just 'Methods' and remove the 'Reagents and Tools Table' form the Methods section of the main manuscript text file. This should only be uploaded separately. Please remove the instructions from the final Reagents and Tools

Table.

Response: In this revision, the Reagents and Tools Table in the Methods section has been removed, and the title also has been revised according to the suggestions.

- Please add the information provided in the EV tables (Table EV1 and EV2) to the reagents and tools table and remove the two files. Please update any callouts and/or provide callouts to the Reagents and Tools Table as appropriate.

Response: The manuscript has been revised according to the suggestion, and callouts have been updated.

- There are callouts to Tables S1-S3 and S4 in the manuscript, but no such tables are provided (in the Appendix). Please check, update the callouts or provide these tables.

Response: In this revision, the callouts to Tables S1-S3 and S4 have been revised.

- Please remove the section 'Supplementary Material' from the manuscript.

Response: The Supplementary Material section has been removed in this revision.

- Please remove now the referee token from The Data Availability section (DAS) and make sure that all datasets are public latest on the date of online publication of the study.

Response: The token from the Data Availability section has been removed, and RNA-seq data will be open at 1st Jun 2025.

- Please make sure that all the funding information is also entered into the online submission system and that it is complete and similar to the one in the acknowledgement section of the manuscript text file. Presently, the grant from the National Key R&D Program of China (2020YFA0907200) is missing from the submission system. Please check.

Response: In this revision, all the funding information has been entered into the online submission system including National Key R&D Program of China (2020YFA0907200).

- Please add a paragraph to the methods section (titled 'Biosafety') providing details on where and how biosafety-relevant experiments were performed and that these were approved, and by whom (institution, government).

Response: In this revision, Biosafety has been added in the methods section.

- You indicate in the checklist that you used a select agent (a biological agent or a toxin has the potential to pose a severe threat) in the study (<https://www.selectagents.gov/sat/list.htm>). If this is true, please declare which agents have been used, that security measures have been applied and which ones, and that you have approval for the use (institutional, governmental). If no select agent has been used, please remove that part from the checklist (changing the entry to 'Not Applicable').

Response: The material in this study are not in Biological Agents and Toxins List, and the checklist has been revised.

- During our figure integrity check, we noted a reuse of the rightmost panel in Fig. 1B (gp48) in Appendix Fig. S1 (A10ZJ24 deltagp48). Please check. If this is intentional, please clearly state this in the respective figure legends.

Response: In this revision, the duplicate or misused images have been deleted from Appendix.

- We also noted overlapping features between some of the images in 6C, 7B and Appendix Fig. S7, which might be cause by the rather low image quality. The contrast of most of plaque-formation assays shown in Figs. 6, 7 and S7 is rather low. Could this be improved?

Response: Following your suggestion, the duplicate or misused images have been deleted. About the limited clarity of plaque morphology in Figure 6,7 and S7, it is not caused by low image quality, but resulted from imaging at 3 days post phage infection of *M. tuberculosis*, in which the bacterial lawn remained underdeveloped.

- Moreover, there is an image reuse within Appendix Fig. S1. Panel 'A10ZJ24 deltagp53' and 'A10ZJ24 deltagp54' seem identical. Please check.

Response: The duplicate or misused images have been deleted for this revision.

In addition, I would need from you uploaded separately:

Response: In this submission, we have followed your comments to upload these files separately.

Referee #1:

The manuscript of Li et al on "A conserved phage phosphoesterase enables evasion of host bacterial antiviral immunity" describes the identification of non-essential genes of a temperate mycobacteriophage A10ZJ24. A modified BRED method was first utilized to produce a series of mutant phages with single gene-deletions. All these mutants reproduced efficiently in *M. smegmatis*, however, some showed clear growth defects in *M. bovis* and *M. tuberculosis*, and the one with gene g48 deletion failed to reproduce in *M. tuberculosis*. With well-designed experiments the authors demonstrated that Gp48 is required for phage genome replication in *M. tuberculosis*. Bioinformatics identified Gp48 as a metallophosphoesterase and that activity was

indirectly confirmed by generating selected active-site single-amino acid substitution mutants. Then Gp48 was shown to be responsible of host strain genomic DNA degradation, and thereby inactivate unknown host anti-phage defense systems that were identified by RNA-seq analysis. Overall, the manuscript presents new information on phage strategy to evade host anti-phage defense.

The English usage of the revised manuscript has been improved, however, I still found a few point where the authors need to revise the text:

L36-40. These two sentences are confusing. Perhaps better like this: While the gp48-mutant phage infected and injected normally its DNA into *M. tuberculosis* cells, it was not able to prevent the activation of the bacterial anti-phage genes which inhibited the replication of the mutant phage genomic DNA.

Response: Thanks for the comments. L36-40 has been revised according to the suggestion.

L83. Remove extra left parenthesis sign

Response: Thanks. The extra left parenthesis sign has been removed.

L207-213. The text does not make sense and does not reflect the results in figure 4B. And I found out that it is due to an omission of a single word "not" from the sentence: both A9GX2 gp53 and A3JS8 gp53 could NOT replace A10ZJ24 gp48 to produce clear plaques on the lawns of *M. tuberculosis* H37Ra.

Response: Sorry. L207-213 has been revised according to the suggestion.

L216-7. Revise the text as you produced three replacement mutants.

Response: L216-217 has been revised according to the suggestion.

L223, and see L259. was removed from the conserved motifs > was located outside of the conserved motifs

Response: Thanks. L223 has been revised according to the suggestion.

L246: grow > growth

Response: Thanks. L246 has been revised according to the suggestion.

L247, and elsewhere: is the ATc concentration indeed nanograms/mL, generally it is used as mikrogramms per mL. This comment is also valid for the antibiotic concentrations mentioned in the Mat & Meth section. L427 forward

Response: Thanks. In this revision, we have followed the suggestion to revise the manuscript.

L482. incubated > incubation

Response: Thanks. L482 has been revised according to the suggestion.

L486. The last sentence does not make sense

Response: The last sentence has been removed.

L500. Check the tense of the last sentence

Response: The tense of the last sentence has been removed.

L517. Described previously

Response: L517 has been revised according to the suggestion.

L519-20. Revise the sentence

Response: L519-20 has been revised.

L523-525. The last two sentences need revision

Response: L523-25 has been revised.

L535-537. The sentence needs revision

Response: L535-37 has been revised.

L541. Also this sentence

Response: L541 has been revised.

L547. PH > pH

Response: L547 has been revised according to the suggestion.

L558. Briefly, linear DNA was ...

Response: L558 has been revised according to the suggestion.

L560. genomic DNA was ...

Response: L560 has been revised according to the suggestion.

L581. What are meant by metabolites?

Response: L581 metabolites has been revised to gene expression.

L586. vectors

Response: L586 has been revised according to the suggestion.

L590-1. The sentence needs revision

Response: L590-1 has been revised.

L593. At what time were the cells collected?

Response: L593 has been revised and given the collection time of cells.

L603-605. The sentence needs revision

Response: L603-605 has been revised.

L785. The sentence is unclear

Response: L785 has been revised.

L833. shadows > highlighting

Response: L833 has been revised according to the suggestion.

L842. SAP is Shrimp alkaline phosphatase, so why give it twice?

Response: L842 SAP has been removed.

L895. sed > used

Response: L895 has been revised according to the suggestion.

Referee #2:

The current revision addresses several of the previous comments and illustrates an interesting interaction between phage genes and antidefense mechanisms which determines host range. A few changes in writing style would make the paper easier to read: In several places the authors have summarized the conclusion of the section in the opening statement. This is confusing to read. eg :Ln 196-213 and 288-300. Additionally, section 288-300 gives the impression that RNAseq of only the gp48 mutant was conducted. It isn't until the last sentence that one realizes that WT RNAseq was also conducted. Rewriting both these sections would improve clarity of the paper. Also rewriting section 301-312 would be beneficial.

Response: Thanks for the comments. In this revision, we have followed the suggestions to modify the paragraphs and improve clarity of the paper.

Ln 257: Why is gp48 toxic to *M. smegmatis*? This is not consistent with the remaining data.

Response: Sorry. We guess something is confusing. The expression of *gp48* is toxic in both *M. smegmatis* and *M. tuberculosis* H37Ra (Fig S5A). Furthermore, its *in vivo* expression can degrade the genomic DNA of both types of mycobacteria (Fig 5C and Fig S5D). The results are very consistent with other data and our model.

Ln 278-282: Why is phage DNA not degraded by purified gp48?

Response: The observation about phage DNA not degraded by purified gp48 is an interesting but the underlying mechanism remains unclear. We hypothesize that mycobacteriophages may have altered their genomic DNA topology similar to previous report in other phages (e.g., through methylation¹ or Z-DNA²). However, the exact mechanism remains to be searched in the future.

Ln 304, Fig 6C : *Mra_2329A* is not seen in figure. Only 1649 and 3122 have an effect. The text is misleading. Similarly Ln 313 and Figure 6D - only 1649 and 3122 have an effect.

Response: Sorry. About this part of the manuscript text has been revised for addressing your concern. The expression of *Mra_1940A* and *Mra_2329A* confers

phage resistance to A10ZJ24 in *M. smegmatis*. And the expression of four genes (*Mra_1649*, *Mra_1940A*, *Mra_2329A* and *Mra_3122*) confers phage resistance to A10ZJ24 in *M. tuberculosis*. Thanks.

Referee #3:

The manuscript suitable for publication in EMBO reports without further revision.

Response: Thanks for the comments.

1. Hausmann R, Gold M. The enzymatic methylation of ribonucleic acid and deoxyribonucleic acid. IX. Deoxyribonucleic acid methylase in bacteriophage-infected *Escherichia coli*. *J Biol Chem*. 1966 May 10;241(9):1985-94. PMID: 5329749.
2. Zhou Y, Xu X, Wei Y, Cheng Y, Guo Y, Khudyakov I, Liu F, He P, Song Z, Li Z, Gao Y, Ang EL, Zhao H, Zhang Y, Zhao S. A widespread pathway for substitution of adenine by diaminopurine in phage genomes. *Science*. 2021 Apr 30;372(6541):512-516. doi: 10.1126/science.abe4882. PMID: 33926954.

Prof. Zheng-Guo He
Wuhan University
115 eastlake road, wuchang District
Hubei 430071
China

Dear Prof. He,

Thank you for the submission of your final revised manuscript to our editorial office. I went through this and your final p-b-p-response, and consider the remaining points of the referees, and the editorial requests, as adequately addressed.

I am thus very pleased to accept your manuscript for publication in the next available issue of EMBO reports. Thank you for your contribution to our journal.

Yours sincerely,
